# Review on Biocompatibility and Prospect Biomedical Applications of Novel Functional Metallic Glasses

**DOI:** 10.3390/jfb13040245

**Published:** 2022-11-16

**Authors:** Michał Biały, Mariusz Hasiak, Amadeusz Łaszcz

**Affiliations:** Department of Mechanics, Materials and Biomedical Engineering, Wrocław University of Science and Technology, Smoluchowskiego 25, 50-370 Wroclaw, Poland

**Keywords:** metallic glass, bulk metallic glass, biocompatibility, in vitro, in vivo, biomedical, Zr-based, Ti-based, mechanical properties, corrosion resistance

## Abstract

The continuous development of novel materials for biomedical applications is resulting in an increasingly better prognosis for patients. The application of more advanced materials relates to fewer complications and a desirable higher percentage of successful treatments. New, innovative materials being considered for biomedical applications are metallic alloys with an amorphous internal structure called metallic glasses. They are currently in a dynamic phase of development both in terms of formulating new chemical compositions and testing their properties in terms of intended biocompatibility. This review article intends to synthesize the latest research results in the field of biocompatible metallic glasses to create a more coherent picture of these materials. It summarizes and discusses the most recent findings in the areas of mechanical properties, corrosion resistance, in vitro cellular studies, antibacterial properties, and in vivo animal studies. Results are collected mainly for the most popular metallic glasses manufactured as thin films, coatings, and in bulk form. Considered materials include alloys based on zirconium and titanium, as well as new promising ones based on magnesium, tantalum, and palladium. From the properties of the examined metallic glasses, possible areas of application and further research directions to fill existing gaps are proposed.

## 1. Introduction

Improving the success rate of surgical procedures along with the life quality of patients after operations is the driving force for the ongoing development of new materials for biomedical applications. The fundamental requirement for such materials is broadly defined biocompatibility. The U.S. Food and Drug Administration (FDA) provided the universal definition of this property as the effect of inducing no measurable harm to the host organism by the material [1]. In this approach, the concept of biocompatibility includes both chemical and mechanical interactions of material with an organism.

One of the widely used standard materials groups for medical implants, surgical tools, and other bio-related devices are metallic alloys, including stainless steels, pure Ti and Ti-based alloys, Co-based alloys, pure Zr and Zr-based alloys, bioresorbable Mg-based alloys, pure Ta, and other miscellaneous [1,2,3]. However, they are often afflicted with many problems including insufficient corrosion fatigue, fretting fatigue resistance, as well as possible toxicity by the release of toxic ions. For the stainless steels and Mg-based materials, corrosion resistance is also problematic. For the stainless steels and Ti-based materials, the wear resistance is a serious concern. Moreover, in the case of implants, the stress shielding effect [4]—which consists of transferring the loads through the stiffer element—is challenging. In the typical case, the load is transferred through the implant (stiffer than the bone) without stimulation of surrounding bones to maintain their properties, which causes bone resorption and implant loosening. Materials such as stainless steels, Co-based materials, pure Ta, and certain alloys from other groups—which are characterized by a Young’s modulus significantly higher than the value for the bones (5.7–18.2 GPa [5])—are particularly prone to triggering this effect. However, it was reported that materials with Young’s modulus values of about 60 GPa are already effective in eliminating the stress shielding effect [6]. Ultimately, the mentioned issues lower the applicational biocompatibility of current materials for biomedical applications. Other problems relate to the inapplicability of some materials for magnetic resonance due to the displacement possibility (e.g., for ferromagnetic materials such as ferritic and martensitic stainless steels [7]) or inducing severe artifacts (e.g., for austenitic stainless steels, Co-based alloys, and even Ti-based alloys [8]). Similarly, artifacts can be induced during X-ray imaging [9].

The efforts to reduce all of the mentioned drawbacks are in constant progress. However, the development limitations in the form of possible alloying compositions, internal structures, and manufacturing techniques are imposed. The possible solution to bypass the disadvantages of the existing material and overcome the imposed limitations is the use of a different group of metallic materials, i.e., Metallic Glasses (MGs) which are characterized by amorphous internal structure.

The majority of metallic alloys at room temperature are characterized by a crystalline structure, which ensures the optimal arrangement of atoms [10]. However, to some extent, it is possible to control the structure at the manufacturing stage by altering the thermodynamic conditions. The structures obtained in rapid cooling processes were noticed in metals as early as in the 1940s [11,12] while applying metal vapors to the cold substrate. The first confirmed formation of an amorphous structure in metals obtained by cooling them from liquid to solid state dates to 1960 [13]. It was produced at that time by the Duwez research team at Caltech through the very rapid cooling of an Au-Si alloy into the shape of thin foil. This new group of metallic materials was named metallic glasses, by similarity to well-known amorphous oxide glasses.

The confirmation of the possibility of changing the internal structure in metallic alloys initiated rapid progress and research on this material group, which is illustrated in Figure 1. This significant development resulted in the discovery of many new compositions of alloys with higher Glass Forming Ability (GFA), the determination of changes in physical and mechanical properties in MGs, and the development of new production methods. Firstly, the MGs were fabricated only in the form of thin ribbons or foils due to the necessity of preserving the critical cooling rate (≥10^6^ K/s) essential to obtain the amorphous internal structure. Later, the optimization of chemical compositions resulted in the opportunity for manufacturing MGs with lower critical cooling rates (≤10^3^ K/s) and greater critical dimensions exceeding 1 mm, so called Bulk Metallic Glasses (BMGs) [14,15], which now can be produced with critical dimensions measured even in centimeters [16,17]. Eventually, since their discovery, many MGs and BMGs compositions based on various elements have been developed. However, the necessity to maintain the critical cooling rate still restricts the maximal, composition-dependent, achievable dimensions.

The non-crystalline internal structure of metallic glasses implies far-reaching changes in physicochemical properties in relation to their crystalline counterparts. This includes most of the crucial mechanical [19,20,21], electrical [22], magnetic [20,21,23] or corrosion parameters [24]. Typically, the results are, among others, increased strength, hardness, toughness, and wear resistance with a lowered Young’s modulus and low plasticity [25,26]. In terms of strength, the MGs are closer to the maximum theoretical value than any other known metallic material [27]. Additionally, they are characterized by increased corrosion resistance [28,29,30,31], as a result of grain boundary absence that reduces the possible corrosion routes and the common presence of alloying elements with the great ability to chemical passivation. This increase is also connected with nearly ideal chemical and structural (including lack of crystalline inclusions) composition homogeneity in the microscale [32], which reduces the number of galvanic micro-cells formed [33]. Moreover, the amorphous internal structure of bulk metallic glasses allows for their thermoplastic processing [34,35,36], which implies the possibility of creating complex shapes in a relatively simple manufacturing process. Ultimately the metallic glasses production methods, usually involving extremely fast heat dissipation, allow for the reduction of the limitation connected with alloying components’ miscibility because the manufacturing conditions are far from equilibrium. It allows for producing multicomponent alloys with a range of pre-designed properties.

With the unique, different than crystalline metallic materials and properties, the MGs and BMGs were already applied or are under thorough consideration for many prospective applications, including biomedical purposes, to form the basis for the next generation of modern devices. In recent times, several metallic glasses were developed specifically for bio-related applications. Figure 1 shows the rapid increase in publications per year concerning metallic glasses in bio-related applications after 2011 and their increasing share in the annual total of publications on metallic glasses. This data shows the potential contained in these materials and the existing perspective for their further use in the biomedical industry.

The rapid development of biocompatible metallic glasses and the multidirectionality of ongoing research create the need for regular organization, comparison, and discussion of varied results. This provides an opportunity to get knowledge of the current state of research and directs future ones. Therefore, this review intends to cover the recent advances in the development, properties, and application possibilities of functional MGs and BMGs, with an emphasis on the promising and most researched Zr- and Ti-based materials devoted to being used in biomedical applications.

## 2. Zr-Based Metallic Glasses

Some of the most promising metallic glasses for biomedical applications are materials based on zirconium, which is characterized by the lack of local and systemic toxicity as well as the formation of stable oxide layers on the surface in biological environments [37]. Moreover, metallic glasses containing zirconium are among the best alloys in terms of glass forming ability [38,39] and are characterized by one of the highest levels of development. This results in the existence of a huge number of defined compositions with determined properties and many of them have been researched or designed specifically to function in biomedical applications. The most recent ones, including bulk materials and thin films, which are characterized by good corrosion resistance in different simulated environments and/or good in vitro/in vivo biocompatibility, are summarized in Table 1 along with their determined mechanical properties.

### 2.1. Mechanical Properties

As presented in Table 1, the exemplary Zr-based metallic glasses currently being researched are characterized by high strengths in the range between 1043 and 1904 MPa, a comparatively low Young’s modulus of 70–127 GPa, and moderate to high hardness reaching 443–843 HV (4.35–9.10 GPa). However, it should be noted, that the higher values of Young’s modulus are obtained mainly for magnetron-sputtered thin films. The possibility of obtaining very high strengths and hardness is perfect both for implants and surgical devices as it ensures the required durability and allows for material mass and volume reduction. For some thin films produced by magnetron sputtering, the observed hardness was even higher than those reported in Table 1 reaching about 1030 HV (11.1 GPa) [56], which enables the possibility of current materials performance improvement by coating application.

Noteworthy, reported in the literature are high values of elastic strain achievable for Zr-based metallic glasses. This parameter is higher than 2% for each material from Table 1 for which data are available. The elastic strain can reach an even remarkable, as for metallic material, value of 6.28% (Figure 2) for the Zr_56_Cu_24_Al_9_Ni_7_Ti_4_ BMG [48]. For comparison in the popular 316 stainless steel, the permissible elastic strain is only 0.34% and in Ti-6Al-4V alloy, 0.67% [57]. The notable increase in elastic strain for metallic glasses makes a significant difference in their interaction with naturally elastic bones. Such high values of elastic strain are very desirable in the aspect of designing bone implants in which elastic deformation should be the default mode with plastic deformation being a sign of a serious issue. The high values of possible elastic strain are also advantageous in the application for stents or other microdevices with changeable shapes, which are delivered to their target destination in compressed form. Some of the materials, including Zr_56_Cu_24_Al_9_Ni_4_Ti_4_Fe_3_ (Figure 2) [48], Zr_60.5_Ti_3_Al_9_Fe_4.5_Cu_23_ [40], Zr_65_Cu_20_Al_10_Fe_5_, Zr_63_Ti_3_Cu_20_Al_10_Fe_5_ [47], and Zr_60+x_Ti_2.5_Al_10_Fe_12.5−x_Cu_10_Ag_5_ (x = 0, 2.5, 5) alloys [49] also exhibit noticeable plastic strain. In this context, it should be noted, that lack of plastic strain is a significant drawback for metallic glasses in certain applications, such as surgical tools, as it causes abrupt failure mode without previous plastic deformation working as a warning sign.

The Young’s modulus of Zr-based metallic glasses (70–127 GPa) is significantly lower than this for the 316L stainless steel (193–210 GPa) and lower even than this for the Ti-6Al-4V alloy (101–125 GPa) [57]. Mentioned higher values of elastic modulus obtained for thin films are of lower importance as in these cases, the Young’s modulus of the substrate materials is the key factor. A low Young’s modulus for Zr-based metallic glasses indicates the enhanced mechanocompatibility with the bone tissue characterized by a Young’s modulus from about 5.7 to 18.2 GPa depending on the measuring direction (for human femoral cortical bone) [5]. Considering load-bearing orthopedic implants, materials with a Young’s modulus closer to the values for bones cause lower bone resorption in the implant vicinity. It relates to reducing the stress shielding effect (see Section 1). Therefore, this area of applications can benefit from the use of Zr-based metallic glasses with a low Young’s modulus.

Some of the BMGs, namely Zr_40_Ti_15_Cu_10_Ni_10_Be_25_, Zr_50_Ti_5_Cu_10_Ni_10_Be_25_, and Zr_40_Ti_15_Cu_10_Ni_5_Si_5_Be_25_ alloys, were also proved to have excellent tribological characteristics followed by exceptionally high wear resistance [45] illustrated in Figure 3 by very low residual depth after the microscratch test in comparison to reference common materials widely used in biomedical applications. Such characteristics are highly attractive to prevent the excessive wear of implants or surgical tools and reduce the number of debris that could potentially lead to the necessity of implant removal. Moreover, it was also confirmed that these tribological parameters can be even more improved by a relatively simple heat treatment process [58], which further increases the area of possible applications.

### 2.2. Corrosion Resistance

An imperative for non-resorbable implants and surgical tools is excellent corrosion resistance in the body environment strongly connected with a limited release of toxic ions. Meeting this condition paves the way for further in vitro and in vivo research. In recent studies, the corrosion resistance of Zr-based metallic glasses in varied simulated environments was investigated by potentiodynamic polarization tests. The reported environments included:Phosphate Buffered Saline (PBS) [32,40,43,47,49,50,51,52,53,54],Simulated Body Fluid (SBF) [41,45,48,59],Artificial Blood Plasma Solution (ABP) [50,51,52],Artificial Saliva Solution (ASS) [50,51,52],Hank’s Balanced Saline Solution (HBSS) [50,51,52,55,60],Ringer’s solution [31],

and standard environments of 0.6 mol/dm^3^ NaCl, 1 mol/dm^3^ HCl, 1 mol/dm^3^ H_2_SO_4_ [44] and 0.1 mol/dm^3^ NaCl combined with 10g/dm^3^ lactic acid [46]. The identified corrosion protection mechanism is connected with oxides layers formation on the surface of the Zr-based materials [30,32,61,62,63,64]. Cumulatively, the research showed excellent corrosion resistance of Zr-based metallic glasses, typically better than for pure Ti and Ti-based alloys. It was expressed by the low value of corrosion current density, as depicted in Figure 4. However, their passive region width often tends to be low [47] for example in comparison with crystalline Ti-based alloys [40,43,45,46,48,49], which can result in pitting corrosion and is visible in Figure 4 as a sudden increase in current density with more positive polarization. The susceptibility of Zr-based metallic glasses to pitting corrosion in chloride-containing solutions [64], such as body fluids, is one of the main concerns related to their corrosion resistance aspect. As human cell membrane potential can reach about 0.1 V, it can stimulate the electrochemical corrosion of implants with low pitting corrosion resistance [65,66,67].

The corrosion resistance is strongly dependent on the chemical composition of the particular alloy and even single percentage variations can alter the alloy behavior creating one of the possible areas for improvement of pitting corrosion resistance. Furthermore, it should be noted that the alloying elements’ influence is not uniform in every composition and the synergistic effect must always be taken into consideration. The popular alloying element addition is Ti, which is known to improve the GFA and mechanical properties considering the content of about 2–3 at.% [40,47] in the ZrTiAlFeCu alloys. However, the optimal content of Ti to enhance the pitting corrosion resistance is much higher (about 4–6 at.%) [47]. For alloys with lower Ti content, this results in low width of the passive region during polarization tests that lead to a faster breakdown in the passive layer [40,47]. It was also reported by Jin et al. [44] that Hf addition at the expense of copper in ZrTiHfCuAl alloys group improves their GFA and mechanical properties or corrosion resistance depending on the concentration of the elements. Still, it is possible to find a composition providing a good balance between the aforementioned parameters. Similar conclusions were drawn by Shi et al. [48] for ZrCuAlNiTiFe alloys, where the adjusted addition of 3 at.% of Fe with reducing the content of Ni improved both the GFA, mechanical properties, and pitting corrosion resistance (Figure 4). Ultimately, the Zr_45_Ti_36_Fe_11_Al_8_ showed remarkable corrosion resistance with no susceptibility to pitting corrosion [32]. All presented works show how crucial the optimization of the chemical composition is to design the desired properties. Moreover, the gathered research shows the importance of minor alloying additions in the formulation of the proper glass-forming alloy for biomedical applications.

Based on discussed studies, it is also rational to look for new non-standard alloying elements to improve the functional properties of the BMGs. For example, Jain et al. [51] evaluated the ZrAlFeCu alloy with the addition of dysprosium. The alloy showed good corrosion resistance due to the addition of one more element promoting the formation of stable oxides. However, the examined alloy was obtained only in the form of a thin ribbon, which currently limits its application possibilities. The exceptional corrosion resistance was also shown for binary ZrPd alloy with the unique nano-granular structure [55]. The applied magnetron sputtering production method gives the prospects to utilize this alloy for thin coatings to gain benefits from the connection of two different materials with complementary properties. By using this technique, there is also a confirmed possibility of obtaining thin films of Zr-based metallic glasses with the controlled addition of nitrogen (ZrCuAlAgN) which also seems to be beneficial for corrosion resistance [56]. For the BMGs, it was recently reported that the pitting corrosion resistance can be improved by introducing very low levels (about 1200 ppm) of oxygen impurity (Figure 5), which causes shrinking of the free volume highly active regions through enabling the structural ordering and without inducing undesirable crystallization [68].

Inevitably, the passive region width increases with the time of exposure to the corrosion environment as the thicker oxides layer is formed which was, for example, shown for the ZrTiCuNiBe materials group [45]. This suggests the possibility of artificial, chemical, or thermal, passivation of the metallic glasses before intended use to improve their corrosion resistance without compositional modifications. Sawyer et al. [69] incorporated a technique of thermal Ceramic Conversion Treatment (CCT) to obtain a thick ceramic passive layer of oxides on the Zr_44_Ti_11_Cu_10_Ni_10_Be_25_ BMG surface during the oxidizing between the glass transition temperature and crystallization temperature. With the right choice of temperatures and times, the technique proved to be effective in increasing the surface hardness (to even 18.32 GPa) and improving the tribological characteristics. It also greatly enhanced the corrosion resistance in terms of the width of the passive region, as the sample was passivated through the entire test up to the polarization of about +1.2 V in relation to corrosion potential. However, the corrosion current density without external polarization was slightly higher for the treated samples indicating a little bit worse corrosion resistance in steady conditions. Nearly all the obtained characteristics are very beneficial for biomedical applications and the method seems to be very promising to test on other MGs compositions. Similar research, but including the temperatures after the crystallization temperature, was reported by Wang et al. [31] for the Zr_56_Al_16_Co_28_ BMG. This study confirmed the widening of the passive region by heat treatment but also indicated the undesirable increase in corrosion current density and revealed the worsening of corrosion resistance in the presence of crystalline phases. The obtained annealing results, in addition to the passivation of an outer layer of materials, should also be interpreted in the context of the thermally activated relaxation process. A reduction of free volumes from the manufacturing stages and associated lower surface electrochemical activity were proven to also be responsible for increased pitting corrosion resistance [70,71].

What is more, recent research confirmed the possibility of increasing the corrosion resistance by surface nano- and micro-structuring by the femtosecond laser [72]. This effect is connected with surface self-cleaning and its multi-level structure.

### 2.3. In Vitro Cellular Research

Very recently, many Zr-based glassy alloys have also been assessed in in vitro cellular research concerning their cytotoxicity which determines the next step for the wider application of this group of materials. The research with the use of MC3T3-E1 mouse preosteoblasts cells showed an excellent cytocompatibility of Zr_60+x_Ti_2.5_Al_10_Fe_12.5−x_Cu_10_Ag_5_ (x = 0, 2.5, 5) BMGs [49], Zr_43.3_Cu_27.8_Ni_15.2_Al_9.1_Ti_4.6_ BMG [73], and Zr_61_Ti_2_Cu_25_Al_12_ BMG [41] in comparison with Ti-based alloys and even PEEK polymer in the case of the Zr_61_Ti_2_Cu_25_Al_12_ alloy. The higher cell proliferation activity resulting in faster cell growth on the surface of Zr_60+x_Ti_2.5_Al_10_Fe_12.5−x_Cu_10_Ag_5_ (x = 0, 2.5, 5) BMGs in comparison to pure Zr and Ti-6Al-4V sample is shown in Figure 6a. The proper MC3T3-E1 mouse cells morphology on the surface of Zr_43.3_Cu_27.8_Ni_15.2_Al_9.1_Ti_4.6_ BMG is visible in Figure 6b. The promising results were obtained even despite the Ag content in the first group of alloys, which is known to exhibit significant toxicity in ionic and nanoparticle forms [74,75,76]. However, the Ag ions cytotoxicity is strictly concentration-dependent [77], which can explain the obtained results together with low relative Ag content and low release rate [78].

Comparably excellent results were also obtained for other Ag-containing BMG (Zr_46_(Cu_4.5/5.5_Ag_1/5.5_)_46_Al_8_ in indirect contact with L929 mouse fibroblasts [60]. Research on compositionally very similar Zr_48_Cu_36_Al_8_Ag_8_ MG thin film, with the use of ISO 10993:5 standard, confirmed also low toxicity of this MG to L929 cells in direct contact [79]. Although Li et al. [80] presented contradictory results for the compositionally identical Zr_48_Cu_36_Al_8_Ag_8_ BMG in nearly equal conditions (ISO 10993:5 standard) in both direct and indirect contact. In his research, the tested material showed severe toxicity to L929 cells, what was connected to a high concentration of released metallic ions, such as Cu, Zr, and Ag. Additionally, the research reported by Rajan et al. [59] considering the same Zr_48_Cu_36_Al_8_Ag_8_ alloy but manufactured in the form of thin films by magnetron sputtering, also showed higher cytotoxicity to MC3T3-E1 cells than commercially pure Ti. It should be mentioned that Cu, which although is a vital microelement [1], can also cause cytotoxic effects [80,81] and its release is dependent on the alloy composition [65]. The result from Li et al.’s research on Zr_48_Cu_36_Al_8_Ag_8_ is rather unusual as the metallic glasses are commonly characterized by low degrees of ions release [46,82,83]. Moreover, the first referenced research by Sun et al. [60] on the (Zr_46_(Cu_4.5/5.5_Ag_1/5.5_)_46_Al_8_ BMG confirmed its biocompatibility even in in vivo animal tests (see Section 2.5). This contradiction prompts research into the effect of minor changes in chemical composition on the ions release in more aggressive environments. Moreover, the ambiguity should be fully explored especially bearing in mind the sensitive and responsible alloys’ bio-related application target.

Another popular type of cells used for cytotoxicity verification purposes are human osteosarcoma cells (HOS), especially from the MG-63 cell lines. Vincent et al. [50] showed a little toxicity of Zr_55_Co_30_Ti_15_ metallic glass to HOS cells using ISO 10993:5 standard, especially in comparison with the different Cu-based metallic glass. The research with the use of MG-63 cells also showed no pronounced cytotoxic effects of Zr_62_Cu_22_Al_10_Fe_5_Dy_1_ (according to ISO 10993:5 standard) [51], Zr_37_Co_34_Cu_20_Ti_9_ [52], and Zr_60.14_Cu_22.31_Fe_4.85_Al_9.7_Ag_3_ [84] metallic glasses. However, the relative viability of cells on the surface of the first two alloys was not perfect and oscillated in the range of 73–91% to the control sample, which can relate to proportionally high Cu content [65] as well as higher ions release.

In other research, the human fibroblasts cells from the cell lines CCD-986sk (from skin) and IMR90 (from lungs) were utilized to evaluate Zr_60.5_Ti_3_Al_9_Fe_4.5_Cu_23_ BMG [40] and a group of ZrTiCuNiBe BMGs with the addition of Si [45], respectively. The first investigated alloy performed better in the cell viability test with CCD-986sk fibroblasts than common Ti-6Al-4V material. However, the overall cell viability after 24 h in comparison to the glass control sample was rather poor (65.4 %). The second group of alloys performed significantly better in contact with IMR90 fibroblasts (in tests according to ISO 10993 standard), providing viability in direct contact better than 88% after 24 h. That was the case despite the Ni and Be content, which are normally considered toxic and generally harmful [85,86]. Similarly, good results were obtained for other Ni- and Be-containing BMGs in tests with mouse fibroblasts from the cell line L929 [60]. These results together, show that the surface-formed oxides layer can provide sufficient insulation of the core material from the surrounding environment reducing the ions migration [49,87] in static (without loads) conditions. The research by Ida et al. [46] confirmed that the amount of varied released ions by Ni-containing Zr_70_Ni_16_Cu_6_Al_8_ BMG is negligible in comparison to Ni-containing stainless steel and even pure Ti (Figure 7). Comparable results were obtained by Wang et al. [88] demonstrating that the ions release from Zr_44_Ti_11_Cu_10_Ni_10_Be_25_ (similar as in [45]) in SBF is low, but also very dependent on chemical composition. It was also shown that Ni and Cu ions release is generally higher among varied compositions and more problematic than Be ions release [88]. The effect of insulating oxides can be also referenced to the above-mentioned excellent cytocompatibility of Ag-bearing BMGs, as the insulating oxides layer could have prevented or slowed down the Ag ions’ migration, maintaining their concentration below the cytotoxicity threshold. However, there is serious concern about the durability of all the described oxides layer in dynamic and tribological conditions or in the case of increased corrosion rates which can noticeably rise the ions’ release and lower the viability of the cells [65]. This leads to the current main development direction of Zr-based metallic glasses for bio-related applications, covering the materials free of potentially harmful elements to eliminate even the possibility of their release. This includes mainly Be and Ni, however, adverse effects were also observed for Cu [65,87,89] and can occur for a sufficient concentration of nearly all important Zr-based metallic glasses components [1] (e.g., Fe, Co, or Al). Because of their essentiality, getting rid of them is extremely difficult without worsening GFA, mechanical properties, or corrosion resistance [38,90,91,92,93,94]. Any deterioration of these functional properties can lead to a worse bio-performance for glassy alloys without the aforementioned elements, which are harmful on their own, than for the ones containing these elements. Therefore, the compositional changes should not be done at all costs. It is recommended to try to use the least harmful components, reduce the content of those not considered biocompatible if possible, look for undiscovered alternatives, and monitor the harmful ions’ release to keep their concentration below the safety limits. At the same time, it should be also done without worsening the overall functional properties. Recently, the Zr-based metallic glass reinforced by nitrogen without any toxic elements was synthesized with a formula Zr_45.5_Ti_42_Si_4.4_N_8.1_ [95]. This alloy showed good compatibility with human skeletal muscle cells (in the test according to ISO 10993 standards) but was only synthesized as a thin film which, for now, restricts its applications to coatings.

The cytocompatibility for existing Zr-based alloys can also be improved by various treatments. The abovementioned CCT heat treatment allowed for improving the Zr_44_Ti_11_Cu_10_Ni_10_Be_25_ BMG surface coverage by Saos-2 human osteosarcoma (sarcoma osteogenic) cells by up to 36%, which was associated with the depletion of the outer oxides layer in the Cu and Ni [69]. Moreover, it was demonstrated that surface pattering can be performed for Zr-based BMG [96] to improve cells adhesion and optimize the healing process in the case of implants. In very recent research, the possibility of laser treatment to control the surface parameters and adhesion for Zr-based metallic glasses was discovered [97,98], which also provides the prospects for further improvement of material-cell interaction.

Another kind of research is related to material-blood contact and in this field, the Zr-based glassy alloys have been successfully evaluated. The research on the Zr_56_Al_16_Co_28_ BMG demonstrated its very low (below 0.12%) hemolytic rates indicating its compatibility with erythrocytes [31]. Moreover, the thin films of the Zr_53_Cu_33_Al_9_Ta_5_ material proved that they do not support blood cells’ adhesion and do not cause platelet aggregation (Figure 8) leading to thrombosis [83,99,100], which is essential for the needles or stents. Furthermore, they also do not support the attachment of cancer cells [83] preventing their further division in the position of the implant.

### 2.4. Antibacterial Properties

A desirable characteristic of materials suitable for implants and medical devices is also the ability to prevent bacterial biofilm formation leading to severe complications including the necessity to remove the implant [101,102]. The often antibiotic treatment ineffectiveness makes it advantageous to obtain the antimicrobial characteristic in the material itself or by its surface modification [103]. Particular metal ions, such as Ag, Cu, or Ni, are effective in preventing the colonization by bacteria [104,105,106,107,108]. However, as mentioned earlier, they are also responsible for cytotoxic effects on organisms’ cells. Especially effective is Ag addition because of the low required ions concentration (way below the cytotoxicity level) to obtain the antibacterial activity [109]. Therefore, in general, a proper balance should be maintained between cytotoxic and antimicrobial effects by ensuring that the concentration of released ions will not exceed the organism’s tolerance limit.

As most of the Zr-based metallic glasses contain the elements suppressing the bacteria growth, they are generally expected to have a high ability to prevent bacterial biofilm formation. Recently, the ZrCuTi and ZrCuAg metallic glasses thin films proved their antibacterial properties (according to JIS Z 2801 standard for the second alloy) both by the lethal to bacteria ions release [110,111] and smooth non-wettable surface [82,110,111], which can also be a way to prevent the biofilm formation [102]. As metallic glasses are characterized by an utterly homogenous internal structure without grain boundaries, the achievable surface roughness is exceptionally low [112,113,114,115,116]. Together, this provides an interesting proposition for coatings of surgical tools, stents, fixators, and other biomedical devices in which cells’ assimilation is not desirable. On the other hand, for the application on implants requiring strong bonding with cells, the extremely smooth surface would be a disadvantage.

Excellent results in terms of antimicrobial properties were obtained in tests according to JIS Z 2801 standard for the ZrCuAlAgN thin films. This material exhibited over 99.999% lethality rate in biofilm formation tests for both *Escherichia coli* and *Staphylococcus aureus* [56], as depicted in Figure 9. Strong antibacterial activity was also demonstrated for the similar ZrCuAlAg composition [79]. Moreover, the research according to JIS Z 2081 standard, by Jabed et al. [54] demonstrated the effectiveness of ZrTiAg thin film metallic glass in antibacterial activity against *S. aureus* indicated by a 48 to 97% reduction in colony-forming units in comparison to uncoated Si substrate. The ZrTiAl metallic glass was not so effective due to the lack of antibacterial elements. What should be noted is that the metallic glass thin film synthesized based on the fully non-toxic formula Zr_45.5_Ti_42_Si_4.4_N_8.1_ [95] exhibited non antibacterial activity in the test according to JIS Z 2801 standards which represents a disadvantage of a kind in its applications and prompts consideration of providing this property.

There is little recent research on the biofilm formation on BMGs. From the available literature, the Zr_58.6_Al_15.4_Co_18.2_Cu_7.8_ alloy showed exceptional effectiveness in eradicating *E. coli* with a 99.99% level [43]. In addition, the ZrAlNiCu and ZrAlNiCuY BMGs proved to exhibit an antibacterial effect on *S. aureus*, greater than the Ti-6Al-4V sample, with the first one showing superior effectiveness [117]. The effect was ascribed to the Cu ions release.

Recently, it was shown that the antibacterial performance of metallic glasses can be further improved by surface modifications. Using the femtosecond laser to obtain the nanostructured surface, the antibacterial properties of the ZrCuNiAlTi BMGs were greatly increased, while maintaining cytocompatibility [72,73,118]. Further increase was observed due to the Ag deposition with H_2_O_2_ treatment [73]. Evaluating the method for other alloys with various chemical compositions can open a new set of possibilities for materials with previously insufficient properties.

### 2.5. In Vivo Research

In vivo animal research represents the next step in the biomedical characterization of potential candidates for wider use, as in the in vitro test it is not possible to observe the full spectrum of mutual material-organism interactions. To the best knowledge of the authors, the Zr-based metallic glasses were tested only on small animals such as mice, rats, and rabbits.

The cylindrical samples of the (Zr_46_(Cu_4.5/5.5_Ag_1/5.5_)_46_Al_8_ BMG with the previously mentioned controversial cytocompatibility in vitro (see Section 2.3) was implanted, together with Ti-6Al-4V specimen, into the thigh bone of Harbin white rabbits [60] as non-load bearing implants. Despite the discrepancies in in vitro studies, complete bone healing and no inflammatory response were observed up to 12 weeks (endo of the study) arguing for the excellent biocompatibility of the examined material. The other research on BALB/c mice was performed with the use of the as cast and annealed Zr_56_Al_16_Co_28_ BMG samples [31]. For the samples in the as-cast state and after annealing below crystallization temperature no inflammatory response, and no cell dysplasia were revealed after 4 weeks since subcutaneous implantation in the dorsal region of mice. Only inflammation signs were observed for the samples heat-treated above the crystallization temperatures, so partially or fully crystallized. These results confirm the deterioration of cytocompatibility with a transition from amorphous to crystalline internal structure what can be associated with worse corrosion resistance and increased ions release.

In real applications, the implant must be fabricated in a specific shape. The research by Ida et al. [46] showed that there is a possibility of machining the Zr_70_Ni_16_Cu_6_Al_8_ BMG to the shape of a screw which was subsequently inserted into the tibia bone of Wistar rats. The reported test, after 28 days, showed that the implanted BMG promotes much faster bone formation than the comparative implant out of grade 2 pure Ti and that it can be even accelerated by the screw implant preload force. This results in much higher stability of the BMG implant measured by pullout torque and shows its good osseointegration ability defined as the existence of no relative motion between the implant and surrounding bone during loading [119]. What can be concerning is the presence of Ni or Cu in the alloy composition. However, the measured concentrations of each BMG compositional element in the brain, liver, kidney, and lung of rats after the 28 days showed no significant difference with the non-implanted control group [46], confirming the lack of the release of the toxic ions as described in Section 2.3. Comparable results were obtained for Ni-bearing Zr_65_Al_7.5_Ni_10_Cu_17.5_ BMG for which no systemic nor local concentration levels of Cu and Ni were increased after 12 weeks since intramedullary implantation into Wistar rat femoral bone [120].

The most recent in vivo studies by Sun et al. on Sprague-Dawley rats showed that the intramedullary placed, load-bearing Zr_61_Ti_2_Cu_25_Al_12_ BMG implant, depicted in Figure 10a and b, provides better integration with the bone (Figure 10c) and earlier formation of larger blood vessels around (Figure 10d) than reference materials (commercially pure Ti and PEEK polymer) at the same time guaranteeing a lower pain level during recovery [41]. Moreover, the research confirmed no pronounced inflammatory response, no metallic ions accumulation, and marginal Cu release, confirming the studied BMG as an excellent candidate for bio-related applications.

On the other hand, different osseointegration results were obtained for the mentioned Zr_65_Al_7.5_Ni_10_Cu_17.5_ BMG, which showed less bone bonding than Ti-6Al-4V and low surface activity after implantation, despite promoting faster surrounding bone healing [120]. What is a disadvantage for the permanent bone implant can be a functional benefit for temporal devices such as intramedullary nails. Research on magnetron sputtered MG thin films with a composition of Zr_48_Cu_36_Ag_8_Al_8_ also showed that their bone integration is rather poor due to the surface non-wettability and such coating can be used rather to prevent the bone affinity, for example, to internal fixators or bone plates [59]. As the composition is similar to the Zr_46_(Cu_4.5/5.5_Ag_1/5.5_)_46_Al_8_ alloys, which showed good osseointegration, it opens a question: what is the influence of wettability and what is the influence of fine chemical composition differences on the material-cells interaction as specified in Section 2.3?

Further improvement of in vivo performance of Zr-based BMG is also possible. The described femtosecond laser surface nanostructuring (see Section 2.4) also proved its effectiveness in the in vivo research on BALB/c mice. The implantation of the Zr_43.3_Cu_27.8_Ni_15.2_Al_9.1_Ti_4.6_ BMG with a laser-induced periodic surface structure beneath the skin of mice [118] and attachment to the femur bone [73] caused no abnormal blood parameters or tissue fibrosis proving the good biocompatibility. The results were obtained even though the intentional application of *S. aureus* on the implant surface before surgery [73] showing the exceptional ability of the material to prevent peri-implant infections.

Finally, the recent applicational studies on the MG thin films showed that particular compositions (e.g., Zr_53_Cu_33_Al_9_Ta_5_) can be very effective for coatings of medical devices such as syringe needles [99,121]. The film improved the durability, hemocompatibility and reduced the cells’ adhesion which resulted in a less invasive procedure (lower retraction forces depicted in Figure 11) and less probability of causing thrombosis.

### 2.6. Summary

The described properties and functionality of Zr-based metallic glasses give hope for their fast implementation in biomedical applications to reduce the percentage of unsuccessful treatments and discomfort of patients. This is supported by their usual good mechanocompatibility with bones illustrated by a low Young’s modulus. Novel Zr-based metallic glasses also showed high strengths and hardness. They are also characterized by very high elastic strains, however, sometimes without further noticeable plasticity. These materials also possess excellent corrosion resistance in terms of corrosion current density. However, they often lack pitting corrosion resistance. Described materials also exhibited cytocompatibility with various cells better or on the same level as currently used materials. It is connected with low ions release through an insulating oxides layer, and lack of toxicity of main components. However, it was shown that suboptimal composition can lead to faster corrosion and ions release lowering the cells’ viability. For some compositions, hemocompatibility and antibacterial properties were confirmed. Recent animal studies also confirmed a lack of negative response for studied Zr-based metallic glasses after implantation. Their use results in better osseointegration, faster blood vessels formation, and most important, a less painful and faster recovery. Moreover, the treatments to improve the mentioned parameters were also recently reported. However, it should also be noted that varied alloying compositions can exhibit very different behavior despite minor elemental changes.

## 3. Ti-Based Metallic Glasses

Ti-based materials are another group of metallic glasses that has recently gained much attention. This metal and its alloys have been used in the biomedical industry on a wide scale for many years [122,123] due to their nearly perfect cytocompatibility [124], exceptional corrosion resistance (connected with stable TiO_2_ oxides formation) [3], very low achievable Young’s modulus—even lower than for the bones—and excellent osseointegration [122]. Moreover, Ti does not play any biological role in the human body and no toxic effects are observed even after taking large doses of this element [1]. Nevertheless, Ti-based alloys also face problems mainly related to wear resistance and fatigue strength [1,2,3,125,126].

After the discovery of metallic glasses, Ti became one of the most widespread and beneficial elements in their compositional design, often in connection with Zr. Therefore, with its excellent biological properties, Ti has formed the basis of a new group of metallic glasses for biomedical applications. Compositions, involving bulk materials and thin films, recently synthesized and characterized bearing in mind such use are summarized in Table 2 when their mechanical properties are available. The most researched materials are the TiZrCuPd and TiZrCuFeSn compositions with their modifications.

### 3.1. Mechanical Properties

From the data gathered in Table 2, it is visible that Ti-based metallic glasses with various properties-modifying additions are characterized by remarkably high ultimate strengths in the range of 1261–3200 MPa, whereby the lowest values are obtained for materials produced by powder metallurgy methods [141]. These values are much higher than the compressive ultimate stress of 42 to 205 MPa for human femoral cortical bone depending on the direction of measurement. Such high values of ultimate strength are a guarantee of the durability of implants or surgical tools even in extreme situations. However, attention should be paid that the values were obtained from the compression tests and should also be verified by tensile tests [142,143]. Moreover, the possibility of the occurrence of the effect of size dependence of mechanical properties [144] dictates the need to check and confirm the properties for different sample sizes to reduce the error in estimating the properties of the final product.

Simultaneously, the materials are not very brittle and generally possess distinctive plastic strain exemplified by Ti_40_Zr_10_Cu_36_Pd_14_ BMG compression test results in Figure 12. The few available values of elastic strain are about 2% (e.g., Figure 12) as in the case of Zr-based metallic glasses, which also means their high capability for elastic deformations, so desirable for load-bearing implants (Section 2.1). The values are approximately six times higher than those for the 316L stainless steel and three times higher than those for the Ti-6Al-4V alloy, which belongs to the current generation of materials for biomedical applications. The hardness is very uniform between varied materials falling within the range of 454–734 HV (4.9 to 7.2 GPa) with the majority of the result between 500 and 600 HV. Additionally, it was demonstrated that the hardness values can be increased by heat treatment near the glass transition temperature and decreased by cryogenic cycling [145].

Similar uniformity is visible for values of Young’s modulus, which ranges between 79.7 and 174 GPa with most values being about 90 to 100 GPa. On average, these are slightly higher than for the Zr-based metallic glasses (Section 2.1) and higher than the lowest achievable for crystalline Ti-based alloys [122]. However, these values are still lower than those for the most popular Ti-6Al-4V alloy (101–125 GPa [57]) and more than two times lower than those for the 316L stainless steel or CoCrMo alloys, which are extensively used in implantology [1], what implies a better efficiency in reducing the stress shielding effect.

The interesting and promising opportunity demonstrated for Ti-based metallic glasses is the possibility of adjusting their Young’s modulus through powder metallurgy. The first research showed that by using amorphous argon atomized powders in hot pressing with Cu particles soluble in HNO_3_ [141] or spark plasma sintering with NaCl particles soluble in water [130], it is possible to obtain the materials with controlled porosity, as seen in Figure 13a. Other methods involve manipulating the porosity only by pressure [146] or temperature [147] control during the spark plasma sintering of mechanically alloyed or argon atomized powders, respectively. By these processes, the Young’s modulus can be controlled (Figure 13b) reaching values of below 10 GPa for the Ti_45_Zr_10_Cu_31_Pd_10_Sn_4_ [130] and Ti_42_Zr_35_Ta_3_Si_5_Co_12.5_Sn_2.5_ [141] alloys with 50% or more porosity or of about 28 Gpa for the Zr_65.5_Fe_22.2_Si_12_ alloy with 26.2% porosity [146]. This manufacturing method also reduces the limitation connected with the necessity to maintain the critical cooling rate as the powder with an already amorphous internal structure is used for further fabrication. However, the pronounced disadvantage of this technology is the simultaneous reduction in yield and ultimate strength with increasing porosity (Figure 13c) even to values lower than for human bones. Although by the proper selection of porosity there is a possibility of obtaining the material with a Young’s modulus comparable to the bone and sufficient strength, which can be the breakthrough in the field of metallic bone implants eliminating the stress shielding effect.

What is important, the research by Hua et al. demonstrated that the Ti_40_Zr_10_Cu_38_Pd_12_ alloy from the popular group of Ti-based BMGs exhibits superior wear resistance to Ti-6Al-4V alloy (Figure 14) measured by three to four times lower wear rate in dry conditions and PBS corrosive environment [138]. Furthermore, wear resistance can be further improved by high-temperature oxidation treatment [148]. This shows the practical alternative to one of the largest issues of crystalline Ti-based alloys, which is the aforementioned wear resistance. Although, it was also revealed that BMG with a very similar composition of Ti_40_Zr_10_Cu_36_Pd_14_ has the same fatigue limit as the Ti-6Al-4V alloy which, with higher initial strength, results in lower calculated resistance to cyclic fatigue [134]. However, the presence of significant casting defects in the BMG should be noted.

### 3.2. Corrosion Resistance

As crystalline Ti-based alloys already show very good corrosion resistance, the expectations and demands from Ti-based metallic glasses are high. Recently, many different compositions were evaluated in various standard and bio-related environments including:Phosphate Buffered Saline (PBS) [128,131,138,148],Borate Buffered Solution with 0.1 M NaCl (BBS) [137],Simulated Body Fluid (SBF) [140,149],Hank’s Balanced Saline Solution (HBSS) [65,130,132,136,150],3.5 wt.% NaCl [127,145,151],0.9 wt.% NaCl [134,136,152,153].

On this basis, it can be stated that in general, Ti-based metallic glasses possess better corrosion resistance in terms of corrosion current density than commercially pure Ti and better or similar to the reference Ti-6Al-4V alloy, which is presented in Figure 15. Similarly, to Zr-based metallic glasses (Section 2.2) the susceptibility to pitting corrosion in chloride-containing solutions is also noticeable [65,127,128,129,131,132,134,136,138,145,148,151,152]. Still, it is not as pronounced, and the passive regions are wider. In general, Ti-based metallic glasses tend to have better pitting corrosion resistance and wider passive regions than Zr-based materials. However, they are also characterized by slightly higher corrosion current densities in corresponding solutions.

There are compositions such as TiCuZrFeSnSiAgTa [136] or TiCuZrFeSnSiSc [131] with slightly worse corrosion performance in terms of corrosion current density than Ti-6Al-4V alloys in Hank’s solution and PBS, respectively. As the TiCuZrFeSnSiAg composition exhibits a lower corrosion current density in PBS in comparison to Ti-6Al-4V alloy [128,129] (Figure 15), a conclusion could be suggested that the substitution of Ta and Sc for Ag deteriorates its corrosion resistance. However, it should be remembered that, from another perspective, they are beneficial for the optimization of GFA and mechanical properties. Furthermore, for Ta addition, the pitting corrosion resistance improving effect can be noticed [136]. The same effect is observable for the addition of Nb in this group of materials [132].

For the TiZrCuPd BMGs group, worse than for the Ti-6Al-4V alloy corrosion performance, measured by corrosion current density, can be observed in PBS [138]. However, in 0.9 wt.% NaCl solution, better corrosion resistance was observed for the same material group in comparison with Ti-6Al-4V alloy [134]. It was also demonstrated that the boron addition improves the corrosion resistance in this group of BMGs [140]. However, its introduction to the alloy requires a complicated pulsed laser deposition technique, which allows only for the obtaining of thin films. In other research, Kuball et al. showed that the sulfur-containing TiZrCuS BMGs are characterized by a very wide passive region, a similar corrosion resistance as Ti-6Al-4V alloy, and much better than Cu-based Cu_47_Ti_34_Zr_11_Ni_8_ BMG in the BBS [137]. However, the Ti_40_Zr_35_Cu_17_S_8_ sample showed isolated corrosion pits after the polarization test (Figure 16a), in contrast to the Ti_50_Zr_25_Cu_17_S_8_ (Figure 16b), and reference Ti-6Al-4V specimen (Figure 16d). What should be mentioned is the research of Lin et al. [65] on a similar group of TiZrCu amorphous thin films demonstrated that excessive Cu addition can significantly deteriorate the pitting corrosion resistance in HBSS leading to excessive ions release.

In recent research [127], the TiZrBeNiFe BMGs group also showed good corrosion resistance in comparison to 304 stainless steel in 3.5 wt.% NaCl. The TiZrHfBeCuNi BMGs group outperformed even Zr-based Zr_41.2_Ti_13.8_Ni_10_Cu_12.5_Be_22.5_ BMG in the same solution with the best corrosion resistance with full substitution of Cu by Ni [151]. For the TiZrBe group, it was also shown that corrosion resistance, along with strength and plasticity, can be improved by Co addition [154]. These groups are also characterized by very good GFA with the possibility of obtaining critical diameters even above 10 mm. However, as one of the few alloys, they contain potentially harmful Be and Ni elements, the release of which has not been evaluated.

To reduce the usage of toxic components, recently Yüce et al. [153] developed a new type of Ti-based metallic glasses with the TiZrSiGeBSn compositions. The alloys were synthesized in the form of thin ribbons, as their GFA is low because of the lack of currently essential elements to obtain bulk materials. However, they showed remarkably wide passive regions, resulting in pitting corrosion resistance, which can be promising for their further development to reduce this common downside of Ti-based metallic glasses. Excellent results were obtained for thin metallic glass films based on the Ti_60_Nb_15_Zr_10_Si_15_ materials which showed significantly lower corrosion current density than the Ti-6Al-4V alloy used as a substrate [149].

The porous materials fabricated by powder metallurgy out of TiZrCuPdSn metallic glass also proved to have superior corrosion resistance to commercially pure Ti and were similar to Ti-6Al-4V [130], which shows that this manufacturing method does not affect the corrosion behavior negatively.

The corrosion resistance of Ti-based metallic glasses, as for the Zr-based (Section 2.2), can be additionally improved by heat treatment. Lin et al. [148] demonstrated, on the basis of the TiZrCuPd BMG, that annealing before the crystallization temperature, together with surface oxidation, improves the pitting corrosion resistance in PBS through enrichment of the outer layer in Ti and Zr. These results are comparable to the ones obtained by Gu et al. [145] from research on the TiZrBeNi BMG. In his study, the corrosion resistance increases in 3.5 wt.% NaCl was attributed not only to the enrichment but also to free volume reactive sites reduction by material relaxation, similarly as for the effect of oxygen impurities for Zr-based alloys (Section 2.2).

### 3.3. In Vitro Cellular Research

Recent cytocompatibility research for the Ti-based metallic glasses was mainly conducted with the use of the MC3T3-E1 mouse preosteoblasts cell line. The excellent viability of these cells was demonstrated on the surface of the Ti_47_Cu_38−x_Zr_7.5_Fe_2.5_Sn_2_Si_1_Ag_2_Nb_x_ (x = 1, 2) BMGs which outperformed even the reference Ti-6Al-4V material, although the addition of Nb has a slightly negative effect on GFA [132]. Similarly, superb results were obtained for the Ti_47−x_Cu_40_Zr_7.5_Fe_2.5_Sn_2_Si_1_Sc_x_ (x = 0, 2) BMGs which also surpassed the Ti-6Al-4V alloy in the WST-1 proliferation assay after 7 days (Figure 17a), despite their slightly worse corrosion performance than this crystalline Ti-based alloy [131]. Moreover, alloys with the same TiCuZrFeSnSi chemical base and addition of Ag also showed similar or superior performance in relation to the Ti-6Al-4V alloy in MC3T3-E1 cells’ adherence, viability, and WST-1 proliferation assays according to ISO 10993:5 standard [128,129]. Together, it points to the suitability of the BMGs from the TiCuZrFeSnSi group as candidates for biomedical materials. The research confirmed their high GFA and the absence of cytotoxicity despite the Cu or Ag content which can cause toxic effects with their ions release above the tolerance threshold (see Section 2.3). However, Lin et al. confirmed that similarly to Zr-based metallic glasses, the ions release and associated cytotoxicity for Ti-based materials are highly dependent on the chemical composition [65].

The Ti_42_Zr_35_Ta_3_Si_5_Co_12.5_Sn_2.5_ material with induced porosity was also evaluated with the MC3T3-E1 cell line [141]. Regardless of porosity, they showed slight cytotoxic effects according to ISO 10993:5 standard in the less challenging indirect contact test in relation to the control sample, which can be connected to the cytotoxicity of Co [156,157]. As the porous structure could be greatly beneficial for tissues adhesion, tissues-material interface properties, osteogenesis, and angiogenesis (blood vessels formation) [158,159], efforts must be made to enhance the cytocompatibility while maintaining the ability to create the foam structure. The other porous BMGs with a composition of Ti_45_Zr_10_Cu_31_Pd_10_Sn_4_ [160] and an extremely bio-friendly Ti_65.5_Fe_22.2_Si_12_ [146] manufactured by powder metallurgy showed very good, comparable to commercially pure Ti, cytocompatibility with MC3T3-E1 cells, demonstrating that porosity does not significantly affect the viability and proliferation of the cells.

The research on the TiCuZrPd thin films modified by B with the use of MC3T3-E1 cells revealed that they also support the proliferation of cells with proper morphology [140]. Furthermore, the proliferation rates were higher than those for the pure Ti control sample, although the B addition appeared to have a slightly adverse effect on cell proliferation, but still way below the cytotoxicity threshold, confirming the studied metallic glass as a promising candidate for implants and biomedical devices coatings. The TiZrCuPd alloys can also be obtained as BMGs. The Ti_40_Zr_10_Cu_38_Pd_12_ metallic glass in a bulk form exhibited no statistically significant differences in cytocompatibility with Saos-2 human osteosarcoma cells (Figure 17b) in relation to the Ti-6Al-4V sample [155]. The similar BMG composition of Ti_40_Zr_10_Cu_36_Pd_14_ was recently found to have very low ions release dominated by Ti and Cu (Figure 18) which was below the cytotoxicity threshold in artificial saliva solution [133]. It also revealed high cytocompatibility with human gingival fibroblast cells illustrated by higher proliferation activity than on the surface of the Ti-6Al-4V specimen, making it a feasible candidate for dental applications. This research confirmed the earlier results of good cytocompatibility of the Ti_40_Zr_10_Cu_36_Pd_14_ BMG estimated in direct contact with MG-63 human osteosarcoma cells and human dermal fibroblasts HDFa, demonstrated by similar proliferation indicators as for the Ti-6Al-4V alloy, although lower than for the control sample of cell culture polystyrene [134]. It was also in accordance with research by Li et al. which demonstrated excellent viability of L929 cells, according to ISO 10993:5 standard, on the Ti_40_Zr_10_Cu_36_Pd_14_ BMG surface and in the extraction medium [80]. Moreover, the substitution of Cu for Co by creating the Ti_40_Zr_10_Co_36_Pd_14_ alloy did not cause cytotoxic effects on MC3T3-E1 and Saos-2 cells [161], indicating that the release of ions is low. In the described alloys, potentially problematic in long-term use or more corrosive environments, can be the presence of Cu and Pd. The latter, even though bio-friendly, can cause the formation of Pd nanoparticles during the pitting corrosion of TiZrCuPd metallic glasses [162] leading to cytotoxic effects [163].

The exceptional cytocompatibility with Saos-2 cells, according to ISO 10993:5 standard and the very low release of ions were demonstrated for thin films composed of Ti_60_Nb_15_Zr_10_Si_15_ [149]. The viability of cells in the 24 h indirect test in extraction media was significantly better for this metallic glass than for the Ti-6Al-4V alloy and, what is remarkable, even higher than 100% in relation to the tissue culture polystyrene control sample. In direct contact, after a long time, the surface of the Ti_60_Nb_15_Zr_10_Si_15_ sample was fully covered with well-adhering cells, showing the excellent lack of toxicity of this material. The same viability, more than 100 %, was also confirmed for this material in indirect contact with L929 mouse fibroblasts according to ISO 10993:5 standard [149].

As for the Zr-based metallic glasses, also for Ti-based ones, the alloy Ti_40_Zr_25_Ni_12_Cu_3_Be_20_ containing Be and Ni elements, showed low cytotoxicity to L929 cells [60]. With a relative to control sample viability of more than 85% after 7 days, it was only slightly worse than the Ti-6Al-4V and Zr-based BMGs which can be related to the low toxic ions release factors (see Section 2.3). Further studies are necessary on the other Ti-based compositions containing these elements to determine their accurate release and biological impact, as the materials with the addition of these elements are characterized by good GFA and mechanical properties [127,154].

Further improvement in Ti-based metallic glasses’ biological performance can be done. Likewise, to Zr-based alloys (Section 2.3) the use of a femtosecond laser to induce the surface structures at the micro- and nanoscale was shown to be effective in improving the cells’ adhesion rates on the Ti-based metallic glasses [164]. Moreover, the stimulation of oxides creation by laser treatment also has a beneficial influence on cell proliferation activity by improving the oxides layer capabilities to prevent the diffusion of the ions. On the contrary, the physically and electrochemically modified surface to obtain the scratch pattern and the mesh-like pattern, respectively, has no considerable influence on the proliferation, adhesion, and differentiation of cells except for directing their growth [155].

In terms of interactions with blood, the Ti-based metallic glasses also showed superior hemocompatibility. The TiNbZrSi alloy coating was found to be non-hemolytic and, unlike the uncoated Ti-6Al-4V sample, does not exhibit platelet adherence and stimulation leading to thrombus formation [149]. The same results were also obtained for the thin films based on TiCuZrPd alloy with B addition [140].

### 3.4. Antibacterial Properties

There is little recent research on the antimicrobial properties of Ti-based metallic glasses. However, the available ones showed that the porous Ti_45_Zr_10_Cu_31_Pd_10_Sn_4_ BMG is effective in the long term in restricting the growth of *S. aureus* [160], what is illustrated in Figure 19. It was confirmed that induced porosity leads to a higher concentration of Cu ions, providing the antibacterial effect (Section 2.4), while still being below the cytotoxicity threshold of MC3T3-E1 cells.

Other research on Ti_40_Zr_10_Cu_36_Pd_14_ BMG confirmed its antibacterial activity against *Aggregatibacter actinomycetemcomitans*, which is common in the oral environment and responsible for periodontal and peri-implant diseases [133]. Moreover, a significant reduction in multispecies biofilm formation was observed for the examined BMG in comparison to the Ti-6Al-4V sample, which was fully covered in a thick layer of different microorganisms present in human oral flora. The antimicrobial action of BMG was connected with a non-wettable surface preventing bacteria adherence and the presence of Cu in the surface layer, similarly as for Zr-based materials (see Section 2.4), confirming the Ti_40_Zr_10_Cu_36_Pd_14_ BMG as a viable candidate for dental implants.

Based on this and the results for the Zr-based materials, the antimicrobial activity is predicted for other recently researched Ti-based metallic glasses with the content of elements inhibiting bacteria growth such as Cu or Ag, with a non-wettable surface being an additional advantage, however, more research is needed in this direction.

### 3.5. In Vivo Research

In the field of in vivo animal research performed with the Ti-based metallic glasses, Kokubun et al. [139] implanted 1.5 mm in diameter Ti_40_Zr_10_Cu_34_Pd_14_Sn_2_ BMG rods transversely in the femoral bones of Sprague-Dawley rats. After 12 weeks, no inflammatory reaction was observed nor implant loosening. The rods were surrounded by tissue and their bone attachment ratio as well as bonding strength were higher than for the pure Ti control group, indicating excellent osseointegration. Furthermore, no diffusion of ions into tissues nor increased systemic Cu level were observed, confirming the excellent short-term performance of the Ti_40_Zr_10_Cu_34_Pd_14_Sn_2_ BMG.

In other research, Lin et al. [65] applied a series of MGs with compositions of Zr_45_Zr_40_Si_15_, Ti_40_Zr_40_Si_15_Cu_5_, and Ti_45_Zr_20_Cu_35_ for implants placed in the tibias of Sprague-Dawley rats (Figure 20) for three, six and twelve weeks. The low GFA of these alloys forced the use of materials in the form of thin ribbons. All rats fully recovered prior to euthanasia. No inflammatory signs were observed afterward, even for the materials with the highest Cu content. The implants were fully surrounded by new bone tissue, the density of which was slightly higher for Cu-free material, what can be associated with higher ions release from the Cu-bearing materials [65]. However, no necrosis was observed, and the overall short-term performance of all materials was good. Nevertheless, the in vitro test showed that, despite excellent results of cytocompatibility for raw materials, the increased corrosion during the electrochemical corrosion test leads to excessive ions release from the last high-Cu alloy. This lowers the viability of D1 mouse bone marrow stem cells to about 78% in 24 h (mild cytotoxicity according to ISO 10993:5 standard). The effects of this were not evident in in vivo study which can be connected to a lower corrosion rate and body fluid circulation removing the released ions from the implant site [65]. However, the proven probability of excessive ions release because of low pitting corrosion resistance (see Section 3.2) for Ti_45_Zr_20_Cu_35_ MG should be a contraindication to its use, especially in a long-term application.

The newly developed porous samples of the Ti_45_Zr_10_Cu_31_Pd_10_Sn_4_ alloy were also evaluated in vivo on the Sprague-Dawley rats [130]. The rectangular 1.5 mm rods were placed for three months in the diaphysis of rats’ femoral bones. It was demonstrated that the porous BMG has excellent biocompatibility, as no ions migration was observed, and bone integration was the same as for the control Ti implant. The porosity allowed for bone ingrowth and well anchoring. The research reported by Liao et al. [165] confirmed the results with the porous Ti_42_Zr_40_Si_15_Ta_3_ BMG, which also supported the cells’ growth into the porous structure without the distinctive BMG-tissue interface, after implantation in the tibias of the New Zealand white rabbits (Figure 21). No inflammatory reaction was observed 6 months after implantation and the newly formed bone had a density similar to the surrounding bone tissue, indicating the excellent osseo-induction property. The discussed results are a good prognosis for the further development of this group of materials and, together with favorable mechanical properties, confirm the right direction of future development.

Ultimately, in a different kind of research, the intradermal reactivity of saline, as well as cotton seed oil extracts from the Ti_60_Nb_15_Zr_10_Si_15_ amorphous alloy, were tested according to ISO 10993:12 and WPTOX 013 standards, on albino rabbits after injection [149]. No erythema or oedema were observed, and the injected extracts did not exhibit indications of toxicity.

### 3.6. Summary

Based on available research, it can be stated that Ti-based metallic glasses possess very favorable properties for bio-related devices. However, their overall development level is slightly below the one for Zr-based metallic glasses. Due to their mechanical properties, corrosion resistance, hemo- and biocompatibility, they may find application as: bulk materials or coatings for implants with particularly good osseointegration and better than current Ti alloys wear resistance, surgical tools with very high strength, or stents without the proneness to cause platelets aggregation. Not without significance is the proven possibility of controlling their mechanical and structural properties by powder metallurgy to reach the bones’ Young’s modulus and cells’ ingrowth capabilities. Their superior to Zr-based metallic glasses pitting corrosion resistance and often better cytocompatibility is a good sign for prolonged use. As Ti-based metallic glasses also often contain fewer potentially harmful elements, they are good candidates for permanent applications. In spite of the single excellent results, the area of antibacterial properties research and in vivo testing is less developed than in the case of Zr-based amorphous alloys, which have already proven their exceptional performance up to a certain point. However, also in the case of Ti-based metallic glasses, some materials were assessed with no negative in vivo response.

## 4. Other Metallic Glasses

Although Zr-based and Ti-based metallic glasses are currently most developed for biomedical applications, they are by no means the only groups suitable for these purposes. The recent fast development of biocompatible metallic glasses constantly provides new materials with different base elements which are briefly summarized below.

### 4.1. Mg-Based Metallic Glasses

Contrary to all described above materials which are meant to be as corrosion resistant as possible to prevent the ions’ release, toxicity, and device damage, the Mg-based metallic glasses are driven by a different philosophy. According to the third generation of biomaterials guidelines which include the ability of a material to trigger specific cellular response [3], the implanted materials are intended to be temporary structures that do not require further surgical removal, and that support the regeneration processes of the organism’s tissues [1]. For this, they are designed to be degradable to allow the native tissue integration and gradual replacement of implant [1,166]. Such an approach enforces the use of only completely bio-friendly components, such as Mg, which is the essential ingredient in the human body and can be absorbed and removed by the organism without causing adverse effects. Recently, the very dynamic development of this material group can be observed with many very similar compositions [167].

Recently, studied Mg-based metallic glasses mainly belong to MgZnCa alloys with hardness in the range of 200 to 300 HV, a tensile strength of about 90–200 MPa [168] and much higher compression fracture strength of about 350–900 MPa [169,170]. These are much lower parameters than for Zr- or Ti-based materials, however still higher than bones properties [5] and Mg-based metallic glasses crystalline counterparts. The Young’s modulus of Mg-based metallic glasses is also very low, reaching 45 to 50 GPa [169,170], which is excellent for eliminating the stress shielding effect. They are also characterized by a very good elastic strain of 1 to 2% (typical for metallic glasses) and lack of plasticity in compression [169,170] what is also connected with brittleness [171]. Typically, they are characterized by GFA allowing to produce diameters up to 5 mm [167]. These parameters can be further improved by the addition of alloying elements [169,172] or solid particles [170], however, usually by one at a time and in quantities of a few at.% as GFA of Mg-based is very susceptible to even minor compositional changes.

The corrosion resistance of Mg-based metallic glasses is low, typically measured in corrosion rates of about 0.15 to 1.7 mm per year in physiological solutions [168,169] what is although their desirable property. The corrosion resistance is still higher than for crystalline Mg-based alloys [167] what allows for their longer biodegradation times leading to full recovery. The second factor is lower hydrogen accumulation during slower resorption. The H_2_ is released in the Mg reaction with H_2_O and underlies the deterioration of the mechanical properties of the Mg-based implants [169]. However, for materials prepared by powder metallurgy, the corrosion rates can reach astonishing values of 75 mm per year in Hank’s solution [170]. The corrosion rates can be also easily manipulated by alteration of chemical composition [168,173,174] or application of bio-friendly coatings [175] what creates the possibility to pre-design the time of implant degradation in the body.

As it was shown, the Mg-based metallic glasses benefit from the group of crystalline Mg-based alloys showing, i.e., bioresorbability and tissue stimulation ability and from the group of metallic glasses showing i.e., improved mechanical and corrosion parameters.

In the initial in vitro cellular research, the Mg_66_Zn_30_Ca_4−x_Sr_x_ (x = 0, 0.5, 1, 1.5 at.%) BMGs showed good cytocompatibility, defined by no visible deviations in the morphology of the cells, with standard MC3T3-E1 mouse preosteoblasts cells [169]. The quantitative indirect (in extraction medium) cytotoxicity assay of Mg_69_Zn_27_Ca_4_ to MC3T3-E1 cells showed higher viability for Mg-based BMG than for pure Mg with a value of 90% in relation to the control group (no cytotoxic effect) [176]. The viability of rabbit primary osteoblasts in extraction medium from Mg_60_Zn_35_Ca_5_ was statistically the same as for the Ti-6Al-4V alloy for 30 days of incubation (Figure 22) [177]. Up to the 14th day, it was above 80% indicating only slight cytotoxicity of the material according to ISO 10993:5 standard. Moreover, the direct contact cytotoxicity of Mg_66_Zn_29_Ca_5_ to MG-63 human osteosarcoma cells was classified as slight to mild according to ISO 10993:5 standard [178]. The not-perfect in vitro research results can be connected to a high concentration of released ions and associated pH changes [170]. What should be noted is that during in vitro tests the culturing medium is the same for the entire time. In the real application in the organism, there is a flow of body fluids that can remove the released ions from the implant site which contributes to more benign testing conditions [65].

It was also shown, on the example of Mg_68_Zn_28_Ca_4_, that Mg-based metallic glasses can exhibit antibacterial properties due to the Zn ions presence [175]. The antibacterial rates against *Colon bacillus* and *Staphylococcus aureus* were greater than 80% and can be further improved to about 100% by nano-hydroxyapatite with ZnO coating.

Recent in vivo research on New Zealand white rabbits with the use of Mg_69_Zn_27_Ca_4_ bone implants showed the new, well-attached, bone accumulation around the implanted material showing its osteogenic activity with no adverse tissue effects after two months since the operation (Figure 23a–c) [176]. The associated healing effect and bone growth for Mg_69_Zn_27_Ca_4_ were even better than for the popular bone substitute—β-tricalcium phosphate (β-TCP), which is illustrated in Figure 23a–f. Other research [177] on New Zealand white rabbits demonstrated no Mg_60_Zn_35_Ca_5_ BMG implant loosening after 24 weeks since implantation to the rabbit femur. The formation of new bone was promoted at a higher rate than for the PLA (polylactic acid) control sample, without inflammatory signs, and maintaining the proper blood chemical composition. Cumulatively, the results give a picture of a material with a good application perspective for bioresorbable implants. However, more detailed cytotoxicity investigations are required.

### 4.2. Ta-Based Metallic Glasses

Tantalum is one of the highly desirable metals to use in biomedical applications due to its excellent biocompatibility and corrosion resistance connected with good mechanical properties [1]. It is used as a minor alloying element in Zr- or Ti-based metallic glasses. However, Ta-based metallic glasses are difficult to manufacture due to the poor GFA. There are very few results described in the literature. The reported Ta_42_Ni_40_Co_18_ BMG with a 2 mm critical diameter also contains potentially harmful Ni and Co and possesses remarkably high strength (2.7 GPa), and a relatively high Young’s modulus (170 GPa) [179]. This directs its possible application to surgical tools or stents. However, it was reported that the creation of Ta-based TaZrCuAlAg thin film metallic glasses by magnetron co-sputtering is possible [180]. Their hardness is about 7.5 GPa (~695 HV), and the Young’s modulus is 130 to 145 GPa depending on composition indicating a possible application for coatings.

Recently, Lai et al. [181] synthesized TaTiZrSi coating without harmful elements by magnetron co-sputtering. Two coatings with compositions of Ta_57_Ti_17_Zr_15_Si_11_ and Ta_75_Ti_10_Zr_8_Si_7_ were proven to exhibit an exceedingly high hardness of 12.1 (~1121 HV) and 15.5 GPa (1435 HV), respectively. Their Young’s modulus were moderately high with values of 133.7 and 144.5 GPa, respectively. They demonstrated corrosion resistance superior to pure Ti in Hank’s solution. What is remarkable is that completely no signs of pitting corrosion were observed even with remarkably high polarization—more than +2 V in relation to corrosion potential (Figure 24a). Performed in vitro cytocompatibility tests with D1 mouse mesenchymal cells showed that, relative to control, the viability of cells for Ti-based MGs is nearly 100% indicating excellent cytocompatibility (Figure 24b). A higher number of cells was attached to the Ta-based MGs than to pure Ti and pure Ta.

Together, this demonstrates the outstanding biological performance of Ti-based MGs coatings and encourages further testing including in vivo.

### 4.3. Pd-Based Metallic Glasses

Due to the biocompatibility of Pd [182,183] and the remarkably high GFA of compositions in which it is present [17] recent biomedical research also includes materials based on this element. It was shown that Pd_40_Cu_30_Ni_10_P_20_ BMG [184] exhibits a hardness of about 500 HV which is more than for popular biomedical materials such as 316L steel, Ti-6Al-4V alloy, and even CoCrMo alloy. Moreover, it possesses exceptional wear resistance both in dry conditions and corrosive phosphate buffer saline (PBS) which is about 30 to 40 times higher than for Ti-6Al-4V alloy (Figure 25a). Its corrosion resistance measured by corrosion current density is also superior to Ti-6Al-4V, as shown in Figure 25b, indicating lower corrosion rates. All these characteristics are significant for biomedical applications. Yet the disadvantage is the presence of potentially toxic Cu and Ni in the described BMG composition, which can be problematic in further in vitro studies.

The Pd_77.5_Si_16.5_Cu_6_ composition [185] with a smaller number of harmful elements, very recently demonstrated excellent hemocompatibility and thrombogenic resistance preventing the platelets aggregation and activation in comparison to Ti-6Al-4V alloy. This results in the possible application of this BMG for stents or blood-pumping devices with a low risk of causing thrombosis.

These two studies open a new field of possibilities toward applications of Pd-based metallic glasses in bio-related applications further expanding the range of available materials.

## 5. Conclusions

This review attempted to collect and summarize the recent advances in research on biocompatible metallic glasses. The presented results demonstrated their appropriateness as prospective materials for a new generation of biomedical devices with the potential to solve problems related to standard materials. The possibility of their manufacturing in bulk form and as coatings for known materials expands the possible fields of application. However, the small obtainable sizes of these materials are a significant problem in further use. In parallel with the study of material properties, the development of manufacturing and processing methods is necessary. The variety of possible chemical compositions and elemental combinations allows for the creation of alloys with pre-designed properties and functionality. Though, systematic testing of new compositions is very important as they can exhibit very different behavior despite minor elemental changes. Moreover, for all materials, it should be noted that usually, good reported mechanical properties are typically determined in compression tests. Though, metallic glasses show asymmetry in tension-compression properties [143]. Therefore, before future use, they need to be evaluated under load conditions more closely resembling real applications. There is also a significant lack in the field of fatigue resistance testing for metallic glasses to be applied in biomedicine. This parameter is essential for the long-term performance of load-bearing implants or surgical tools.

Recent research on Zr-based metallic glasses showed that their achievable mechanical and biological properties are suitable for prospective bone implants with good osseointegration and significantly reduced stress shielding effect, temporary fixtures with low bone affiliation, stents with a low probability to cause thrombosis, needles causing low tissue damage, or surgical tools with antimicrobial properties and increased lifetime. However, for now, they were assessed only in low time intervals on small animals. Attention should be paid to the long-term performance of Zr-based metallic glasses in the body environment as they often contain potentially toxic essential elements. There is a lack of knowledge about the ions’ release in corrosive environments under a cycling load. After more demanding tests in a longer time and more load-bearing applications with fatigue monitoring, it will be possible to determine their further biomedical efficacy.

Based on collected research it was shown that recent Ti-based metallic glasses exhibit properties particularly desirable for surgical tools such as high strength, lack of brittleness, and wear resistance which is a significant drawback of crystalline Ti-based alloys. In terms of bone implants, promising research direction is also the reduction of the Young’s modulus by introducing porosity. Moreover, it also allows for bone ingrowth and better osseointegration. There is a lack of antimicrobial activity studies of Ti-based metallic glasses, especially since this group of materials often contains fewer elements with such activity. From the cytocompatibility point of view, concerning is the presence of Cu in nearly all recent compositions—especially since its release was not studied in dynamic conditions. Future considerations should also include further in vivo testing of materials with proven in vitro biocompatibility, as this field is not as advanced as for Zr-based metallic glasses.

The common point between Zr- and Ti-based metallic glasses is lack of pitting corrosion resistance in chloride containing environments. Further compositional optimizations and treatments development are needed to reduce this drawback with maintaining other properties. A good starting point are those methods for which efficacy has been proven for the selected materials.

As even the best synthetic material for biomedical applications is not able to self-heal like bones, the development of bioresorbable Mg-based metallic glasses seems to be the rational direction. These materials only support natural tissue regeneration processes, possess comparable properties to bones, and are composed of elements common in the human body. However, recent in vitro research also showed varied cytotoxicity of current materials due to the high ions concentration and pH changes. For future consideration is if it has an impact on in vivo performance.

The new group of Ta-based metallic glasses showed very favorable corrosion properties and cytocompatibility. The biggest drawback to solve is the very inferior glass forming ability. In addition, the achievable Young’s modulus are too high to avoid the stress shielding effect in bone implants. However, other applications are viable after more detailed biological studies which are a novel and open field for these materials.

Ultimately, the well-known Pd-based materials also recently showed promising initial results regarding corrosion and wear resistance, as well as hemocompatibility. Further cytocompatibility studies are necessary to assess their usability as the presence of Cu and Ni is concerning.

## Figures and Tables

**Figure 1 jfb-13-00245-f001:**
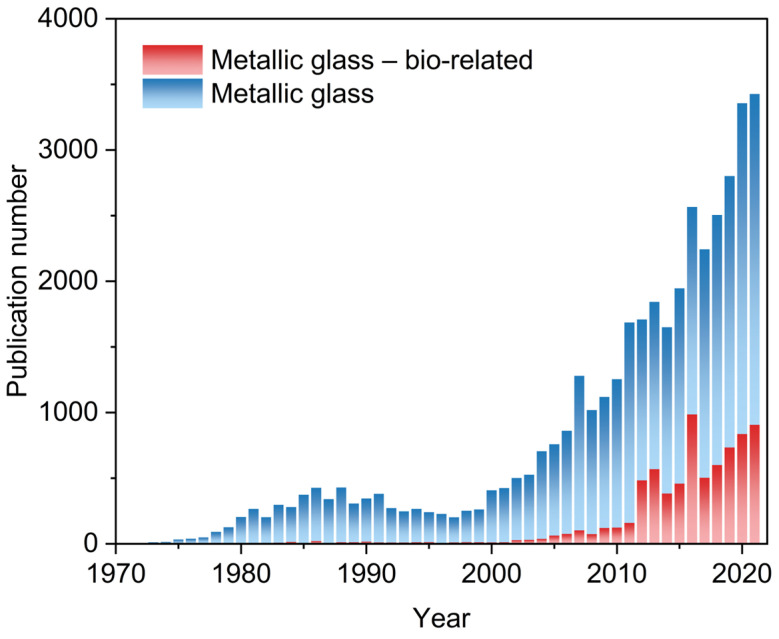
The number of publications per year related to metallic glasses from 1973 to 2021 with the corresponding number of publications per year related to metallic glasses for bio-related applications according to Dimensions tool by Digital Science & Research Solutions, Inc. [18].

**Figure 2 jfb-13-00245-f002:**
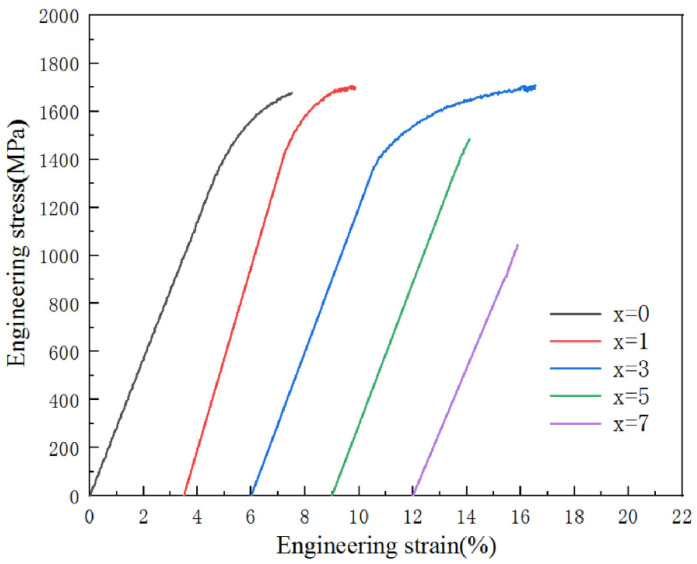
Stress-strain curves obtained in uniaxial compression test for the as-cast Zr_56_Cu_24_Al_9_Ni_7-x_Ti_4_Fe_x_ (x = 0, 1, 3, 5, 7 at.%) alloys. Materials were fully amorphous for x = 0, 1, 3, and partially amorphous for x = 5, 7. (Reprinted with permission from ref. [48]. Copyright © 2022 Elsevier B.V., Amsterdam, The Netherlands).

**Figure 3 jfb-13-00245-f003:**
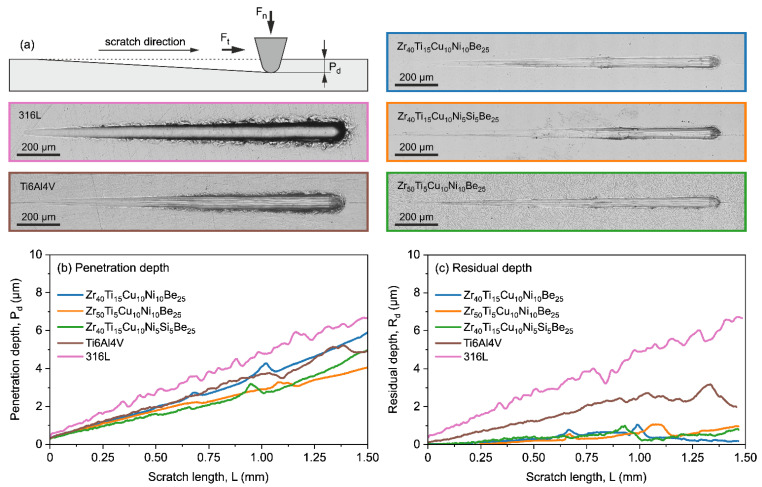
Results of microscratch tests recorded with a gradually increasing load from 0.5 to 10 N with a scratch speed of 1 mm/min and pre- and post-scan measurements with the load of 0.03 N. Images of scratches (**a**) and corresponding scratch parameters such as penetration depth (**b**) and residual depth (**c**) for the Zr_40_Ti_15_Cu_10_Ni_10_Be_25_, Zr_50_Ti_5_Cu_10_Ni_10_Be_25_, Zr_40_Ti_15_Cu_10_Ni_5_Si5Be_25_ alloys, reference 316L surgical steel and Ti6Al4V alloy obtained in scratch tests. (Reprinted and adapted from ref. [45] under Creative Commons CC-BY license).

**Figure 4 jfb-13-00245-f004:**
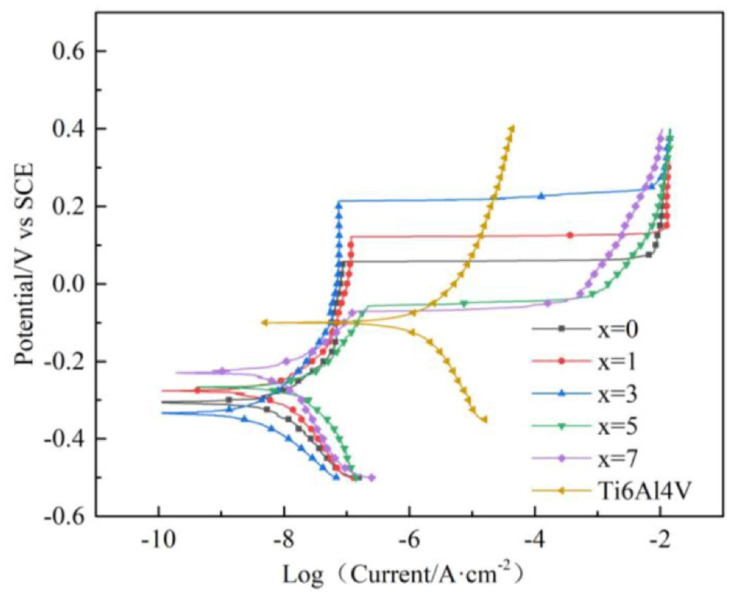
Typical polarization curves for Zr-based metallic glass in comparison to Ti-6Al-4V alloy on the example of Zr_56_Cu_24_Al_9_Ni_7−x_Ti_4_Fe_x_ (x = 0, 1, 3, 5, 7 at.%) in SBF solution at 35.7 °C. (Reprinted with permission from ref. [48]. Copyright © 2022 Elsevier B.V., Amsterdam, The Netherlands).

**Figure 5 jfb-13-00245-f005:**
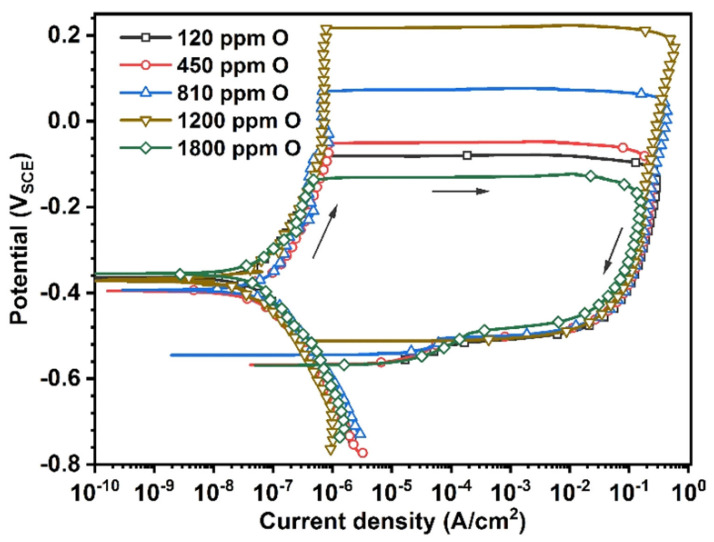
Polarization curves obtained in potentiodynamic test in 3.5 wt% NaCl environment for Zr_61_Ti_2_Cu_25_Al_12_ BMG with different oxygen impurity. (Reprinted with permission from ref. [68]. Copyright © 2021 Elsevier B.V., Amsterdam, The Netherlands).

**Figure 6 jfb-13-00245-f006:**
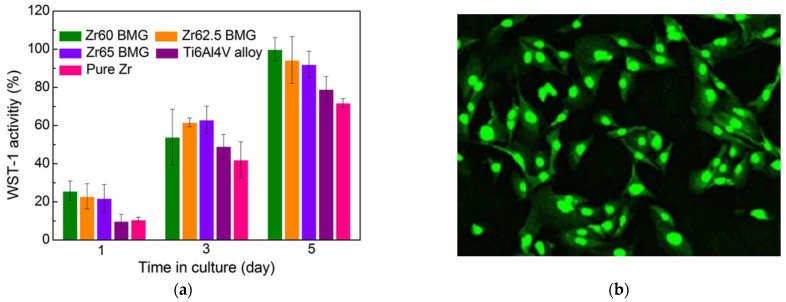
(**a**) WST-1 assay protocol results for the Zr_60+x_Ti_2.5_Al_10_Fe_12.5−x_Cu_10_Ag_5_ (x = 0, 2.5, 5) BMGs and reference pure Zr and Ti-6Al-4V alloy, illustrating proliferation activities of MC3T3-E1 cells after a specific time. Results are given as a percent of the highest value. (Reprinted with permission from ref. [49]. Copyright © 2014 Elsevier B.V., Amsterdam, The Netherlands); (**b**) Fluorescence stained MC3T3-E1 cells on the polished surface of Zr_43.3_Cu_27.8_Ni_15.2_Al_9.1_Ti_4.6_ BMG after 14 days of incubation. No visible changes in cell morphology are observed. (Reprinted with permission from ref. [73]. Copyright © 2022 American Chemical Society, Washington, DC, USA).

**Figure 7 jfb-13-00245-f007:**
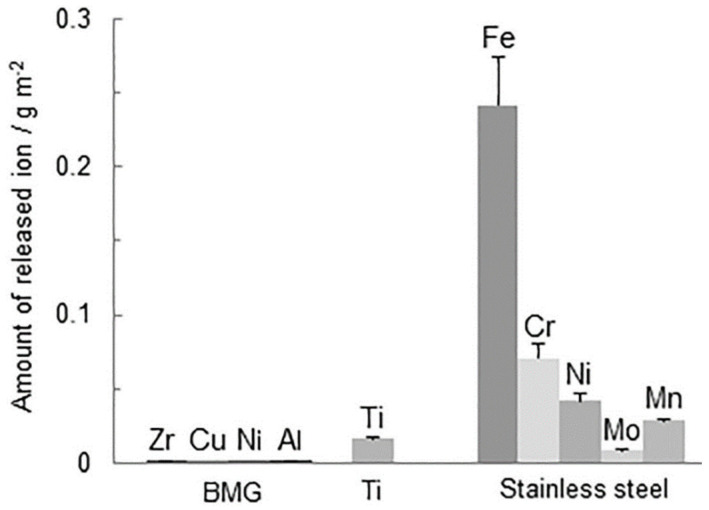
Amounts of ions released from the Zr_70_Ni_16_Cu_6_Al_8_ BMG, 316L stainless steel, and grade 2 pure Ti in 0.6 mol/dm^3^ NaCl, 1 mol/dm^3^ HCl, and 1 mol/dm^3^ H_2_SO_4_ solution after 7 days in 37 °C. (Reprinted with permission from ref. [46]. Copyright © 2018 Elsevier B.V., Amsterdam, The Netherlands).

**Figure 8 jfb-13-00245-f008:**
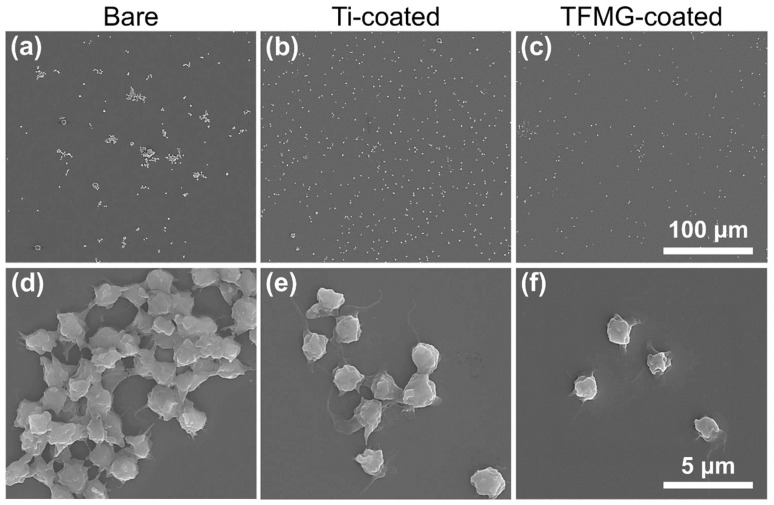
SEM images of human blood platelets on (**a**,**d**) bare glass, (**b**,**e**) Ti-coated glass, (**c**,**f**) glass coated with a thin film of Zr_53_Cu_33_Al_9_Ta_5_ metallic glass. Images (**a**–**c**) represent low magnification, and images (**d**–**f**) high magnification. (Reprinted with permission from ref. [83]. Copyright © 2018 Elsevier B.V., Amsterdam, The Netherlands).

**Figure 9 jfb-13-00245-f009:**
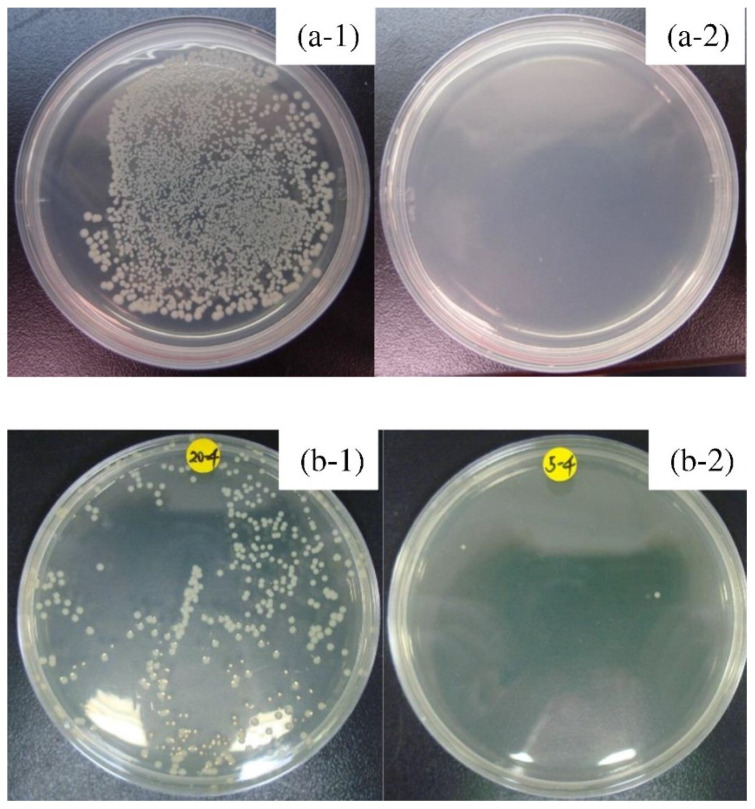
(**a**) Escherichia coli colonies on the dishes corresponding to (**a-1**) 304 stainless steel and (**a-2**) Zr_40.6_Cu_41.5_Al_14_Ag_4_ MG thin film; (**b**) Staphylococcus aureus colonies on the dishes corresponding to (**b-1**) 304 stainless steel, and (**b-2**) Zr_40.6_Cu_41.5_Al_14_Ag_4_ MG thin film. Test according to JIS Z 2801 standard. (Reprinted with permission from ref. [56]. Copyright © 2017 Elsevier B.V., Amsterdam, The Netherlands).

**Figure 10 jfb-13-00245-f010:**
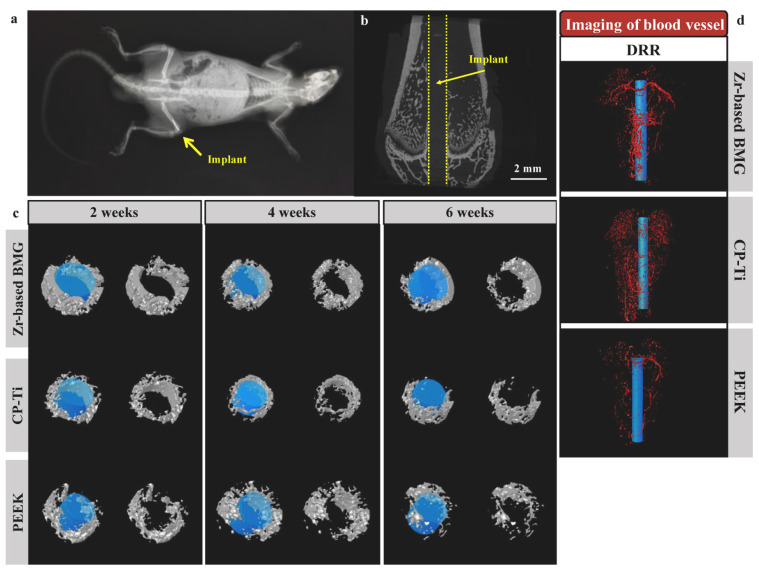
(**a**) X-ray image of Sprague-Dawley rat with the Zr_61_Ti_2_Cu_25_Al_12_ BMG implant; (**b**) Reconstructed image of bone with the intramedullary placed implant; (**c**) 3D reconstructions showing the bone surrounding the implants out of the Zr_61_Ti_2_Cu_25_Al_12_ BMG, commercially pure Ti and poly-ether-ether-ketone (PEEK) after a specific time; (**d**) Digitally micro-angiography reconstructed radiograph (DRR) of blood vessels in the vicinity of implants out of different materials. (Reprinted and adapted from ref. [41] under Creative Commons CC-BY license).

**Figure 11 jfb-13-00245-f011:**
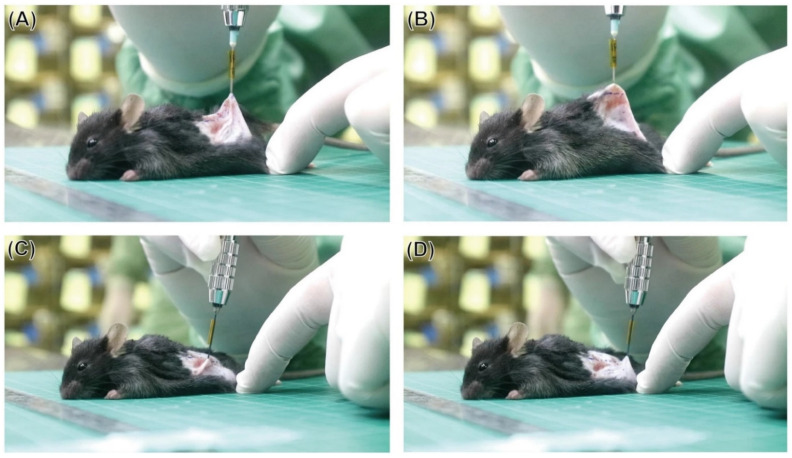
Images of needles retraction after puncturing the mouse dorsal skin for (**A**,**B**) bare stainless steel hypodermic needles, and (**C**,**D**) Zr_53_Cu_33_Al_9_Ta_5_ metallic glass coated hypodermic needles. (Reprinted from ref. [99] under Creative Commons CC-BY license).

**Figure 12 jfb-13-00245-f012:**
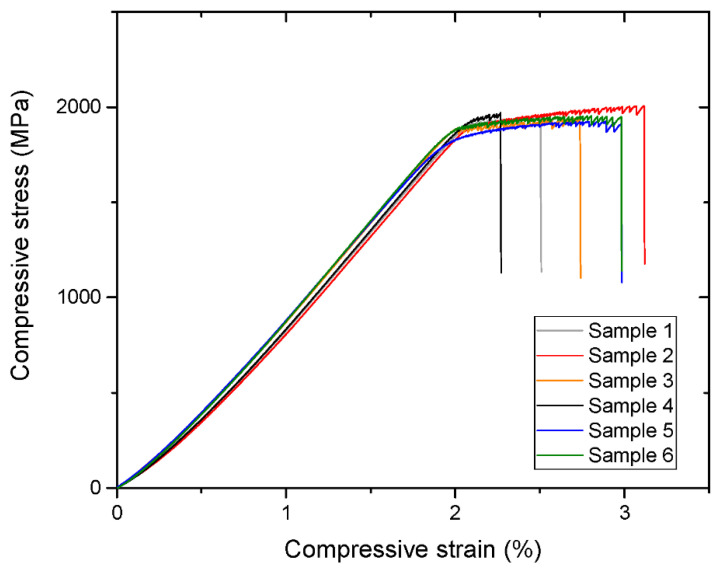
Stress-strain curves for Ti_40_Zr_10_Cu_36_Pd_14_ BMG obtained in compression tests for samples with a diameter of 3 mm and a height of 5 mm. (Reprinted from ref. [134] under Creative Commons CC-BY license).

**Figure 13 jfb-13-00245-f013:**
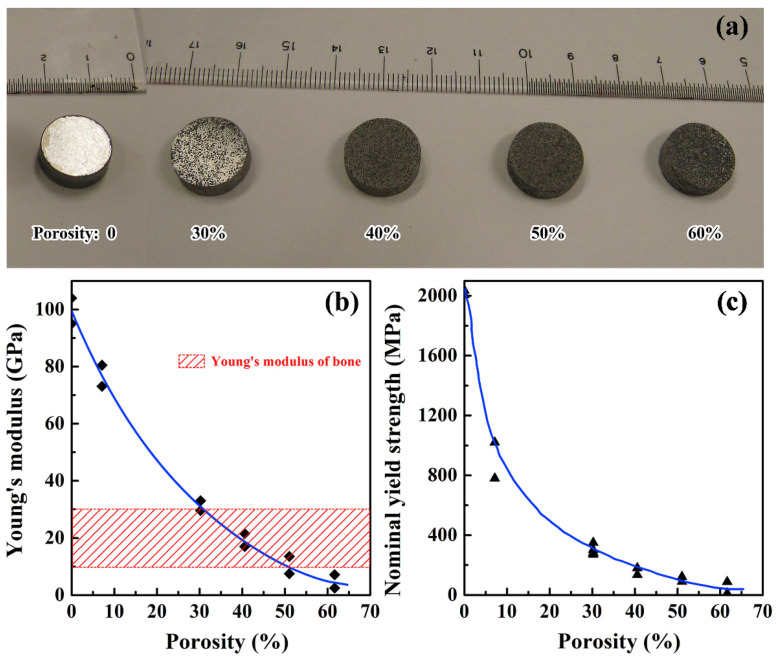
(**a**) The physical appearance of the Ti_45_Zr_10_Cu_31_Pd_10_Sn_4_ BMG with different porosity degrees fabricated by spark plasma sintering; (**b**) Young’s modulus of the Ti_45_Zr_10_Cu_31_Pd_10_Sn_4_ BMG with different porosity percent; (**c**) Nominal yield strength of the BMG with different porosity percent. (Reprinted and adapted from ref. [130]. Copyright © 2017 Elsevier B.V., Amsterdam, The Netherlands).

**Figure 14 jfb-13-00245-f014:**
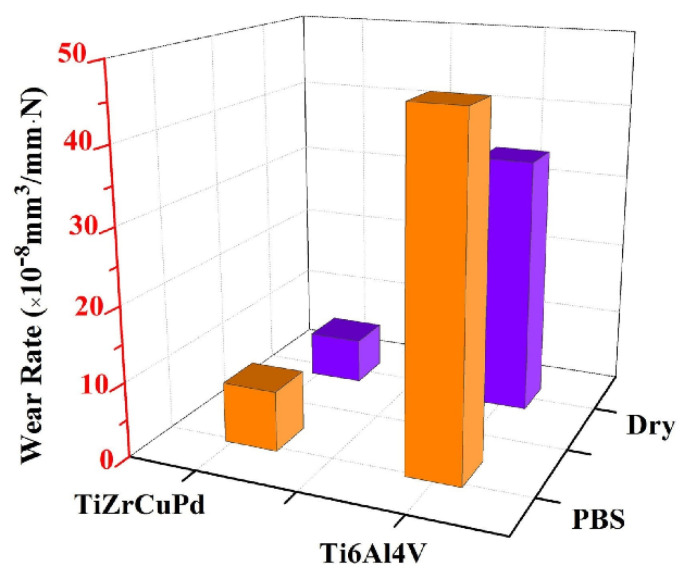
Wear rates of Ti_40_Zr_10_Cu_38_Pd_12_ BMG, and Ti-6Al-4V alloy measured in air, and phosphate buffered solution (PBS) corrosive environment. (Reprinted with permission from ref. [138]. Copyright © 2020 Elsevier B.V., Amsterdam, The Netherlands).

**Figure 15 jfb-13-00245-f015:**
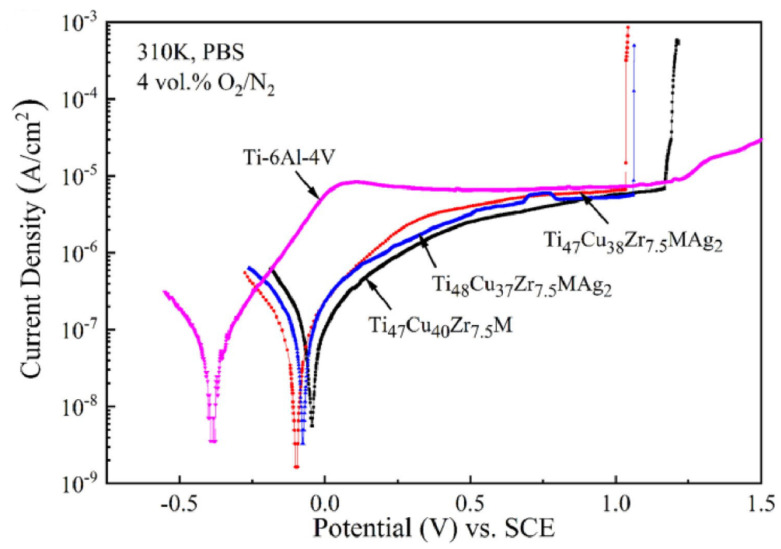
Typical polarization curves for Ti-based metallic glasses in comparison to Ti-6Al-4V alloy on the example of the TiCuZrMAg (M = Fe_2.5_Sn_2_Si_1_) group obtained in PBS solution at about 36.9 °C. (Reprinted with permission from ref. [129]. Copyright © 2021 Elsevier B.V., Amsterdam, The Netherlands).

**Figure 16 jfb-13-00245-f016:**
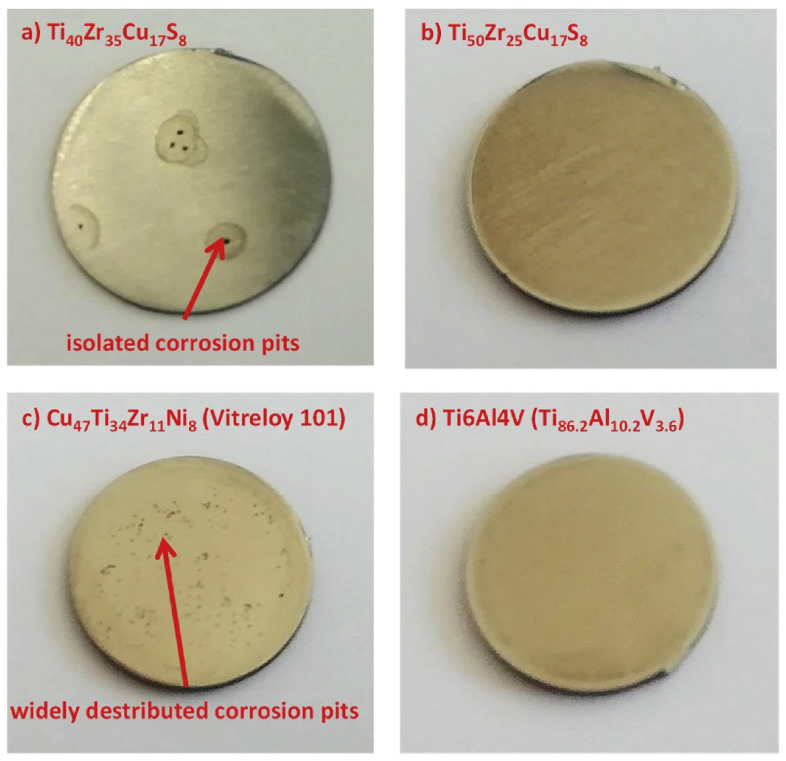
Images of samples surface after the potentiodynamic tests in borate buffered 0.1 M NaCl solution for (**a**) Ti_40_Zr_35_Cu_17_S_8_BMG, (**b**) Ti_50_Zr_25_Cu_17_S_8_ BMG, (**c**) Cu_47_Ti_34_Zr_11_Ni_8_ BMG, and (**d**) Ti-6Al-4V alloy. (Reprinted with permission from ref. [137]. Copyright © 2019 Elsevier B.V., Amsterdam, The Netherlands).

**Figure 17 jfb-13-00245-f017:**
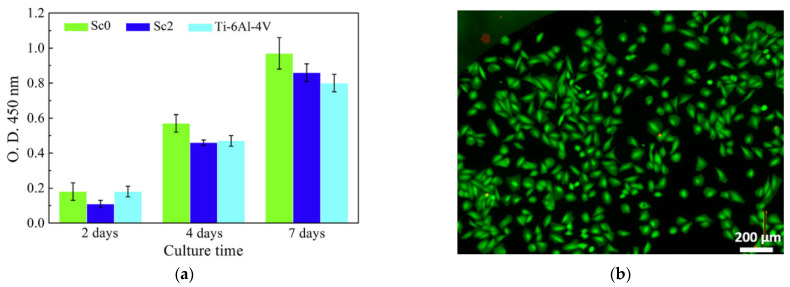
(**a**) Results of the WST-1 assay protocol for the Ti_47−x_Cu_40_Zr_7.5_Fe_2.5_Sn_2_Si_1_Sc_x_ (x = 0, 2) BMGs and the reference Ti-6Al-4V alloy, illustrating the proliferation activities of MC3T3-E1 cells after a specific time. Results were given as 450 nm light absorbance. (Reprinted with permission from ref. [131]. Copyright © 2016 Elsevier B.V., Amsterdam, The Netherlands); (**b**) Fluorescence stained Saos-2 cells on the polished mirror-like surface of Ti_40_Zr_10_Cu_38_Pd_12_ BMG after 24 h of incubation. (Reprinted and adapted from ref. [155] under Creative Commons CC-BY license).

**Figure 18 jfb-13-00245-f018:**
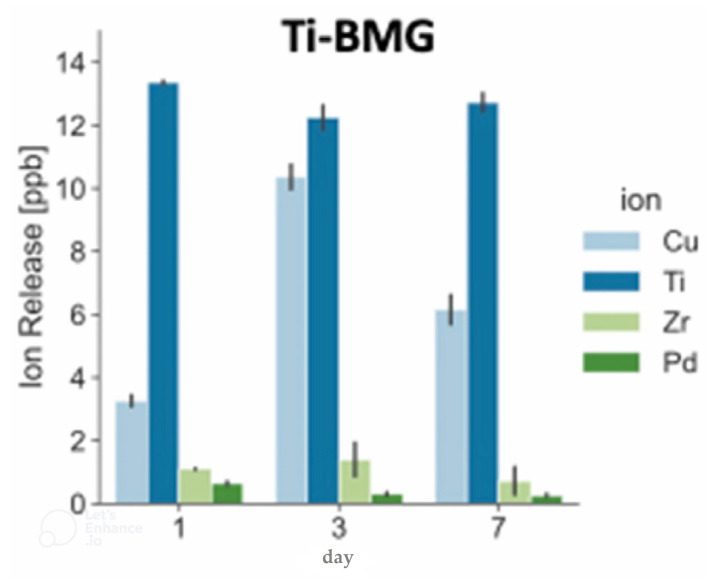
Ions release from Ti_40_Zr_10_Cu_36_Pd_14_ in artificial saliva solution. Measured by inductively coupled plasma mass spectrometry (ICP-MS) after 1, 3, and 7 days. (Reprinted and adapted from ref. [133] under Creative Commons CC-BY license).

**Figure 19 jfb-13-00245-f019:**
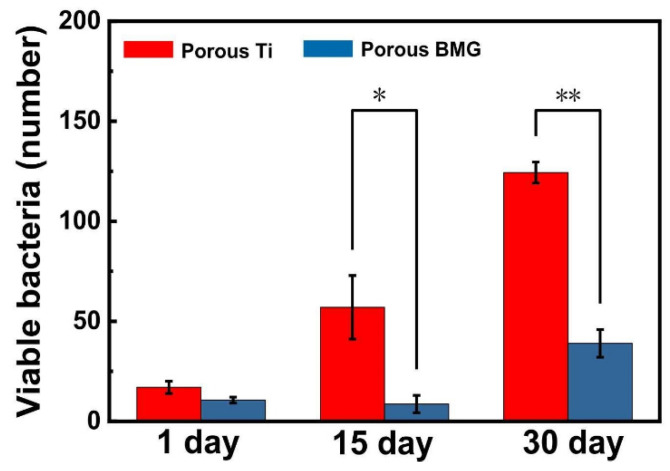
The number of *S. aureus* bacterial colonies co-cultured with the porous Ti_45_Zr_10_Cu_31_Pd_10_Sn_4_ BMG and porous pure Ti as a reference. The *p*-value is <0.01 for * and <0.001 for **. (Reprinted with permission from ref. [160]. Copyright © 2022 Elsevier B.V., Amsterdam, The Netherlands).

**Figure 20 jfb-13-00245-f020:**
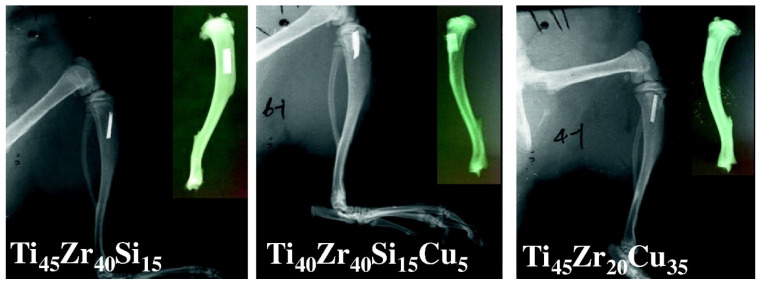
X-ray image of the Sprague-Dawley rat with MG tibia implants of the Ti_45_Zr_40_Si_15_, Ti_40_Zr_40_Si_10_Cu_5_, and Ti_45_Zr_20_Cu_35_ alloys. The main images were taken immediately after implantation, the insets were taken after euthanasia after 6 weeks. (Reprinted and adapted with permission from ref. [65]. Copyright © 2017 Elsevier B.V., Amsterdam, The Netherlands).

**Figure 21 jfb-13-00245-f021:**
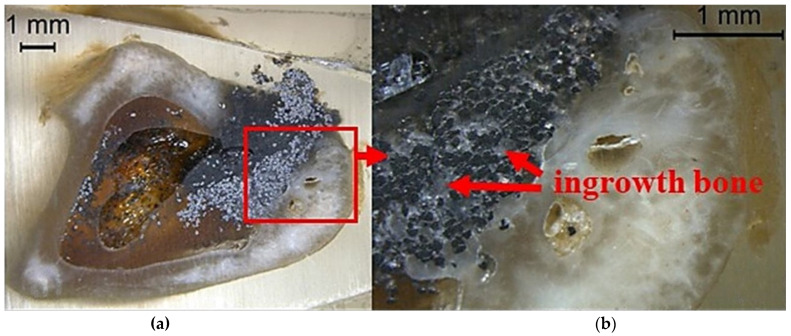
Optical images showing the cross-section of rabbit bone with implanted porous Ti_42_Zr_40_Si_15_Ta_3_ metallic glass sample and bone ingrowths after six months in (**a**) low, and (**b**) high magnification. (Reprinted and adapted from ref. [165] under Creative Commons CC-BY license).

**Figure 22 jfb-13-00245-f022:**
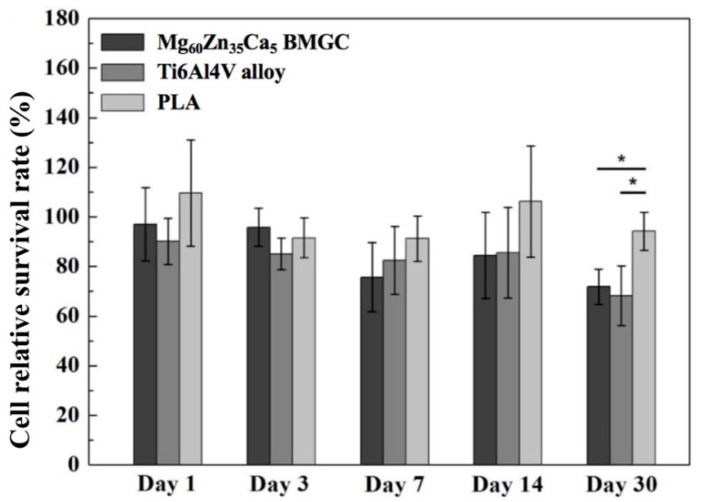
Relative to the control group survival rate of rabbit primary osteoblasts in extraction medium derived after different times from Mg_60_Zn_35_Ca_5_ BMG, Ti-6Al-4V alloy, and PLA polymer. The *p*-value is <0.05 for *. (Reprinted and adapted from ref. [177] under Creative Commons CC-BY license).

**Figure 23 jfb-13-00245-f023:**
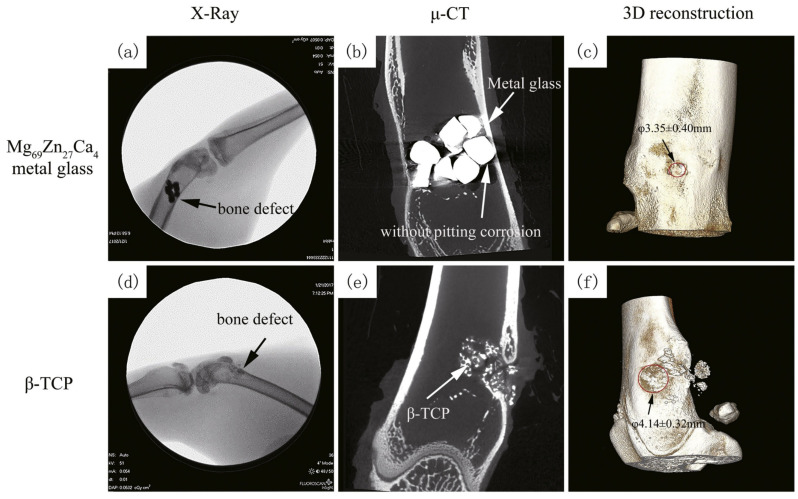
(**a**,**d**) X-ray, (**b**,**e**) micro-computed tomography (μ-CT), and (**c**,**f**) 3D reconstruction images of rabbit bone defects filled with (**a**–**c**) Mg_69_Zn_27_Ca_4_ metallic glass, and (**d**–**f**) β-tricalcium phosphate (β-TCP) after two months since operation. (Reprinted with permission from ref. [176]. Copyright © 2019 Elsevier B.V., Amsterdam, The Netherlands).

**Figure 24 jfb-13-00245-f024:**
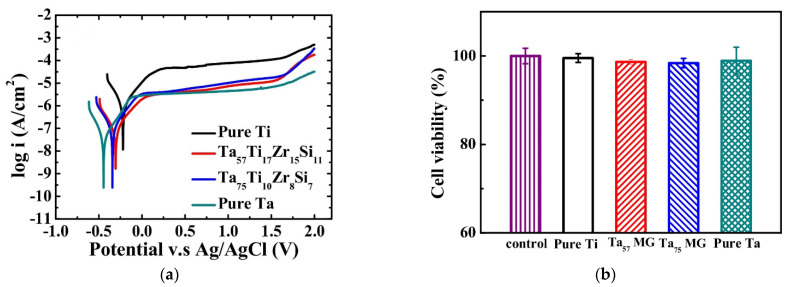
(**a**) Potentiodynamic test in Hank’s Balanced Saline Solution (HBSS) results for pure Ti, pure Ta, Ta_57_Ti_17_Zr_15_Si_11_, and Ta_75_Ti_10_Zr_8_Si_7_ metallic glasses; (**b**) Relative to control group D1 cells viability after 72 h obtained from MTS assay for pure Ti, pure Ta, Ta_57_Ti_17_Zr_15_Si_11_, and Ta_75_Ti_10_Zr_8_Si_7_ metallic glasses. (Reprinted with permission from ref. [181]. Copyright © 2018 Elsevier B.V., Amsterdam, The Netherlands).

**Figure 25 jfb-13-00245-f025:**
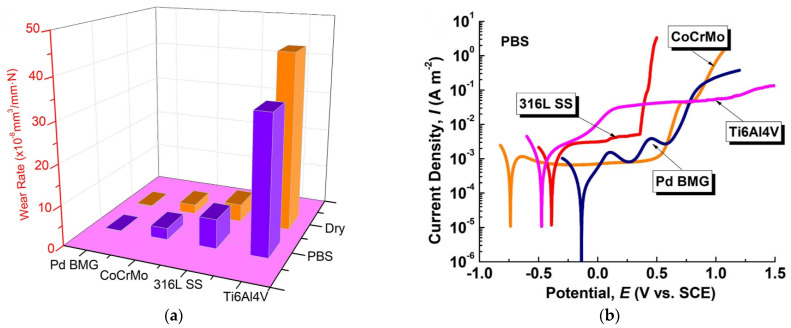
(**a**) Wear rates for Pd_40_Cu_30_Ni_10_P_20_ BMG, CoCrMo alloy, 316L stainless steel, and Ti-6Al-4V alloy in dry conditions and phosphate buffered saline (PBS) solution; (**b**) Polarization curves for Pd_40_Cu_30_Ni_10_P_20_ BMG, CoCrMo alloy, 316L stainless steel, and Ti-6Al-4V alloy obtained in potentiodynamic test in PBS. (Reprinted and adapted with permission from ref. [184]. Copyright © 2020 Elsevier B.V., Amsterdam, The Netherlands).

**Table 1 jfb-13-00245-t001:** Recent exemplary Zr-based amorphous alloys considered for biomedical applications together with their technological details and mechanical parameters: d—obtained diameter/thickness, σ_max_—compressive ultimate strength, E—Young’s modulus, *H*_V_—Vickers hardness, ε_el_—elastic strain.

Composition (at.%)	Production Method	d (mm)	Structure	σ_max_ (MPa)	E (GPa)	*H*_v_ (HV)	ε_el_/ε_pl_ (%)	Ref.
Zr_63.5−x_Ti_x_Al_9_Fe_4.5_Cu_23_ (x = 0, 1.5, 3, 4.5, 6)	Arc melting/suction casting	3–10	Fully amorphous	1580–1690	–	–	–/0.9–4.7	[40]
Zr_61_Ti_2_Cu_25_Al_12_	Arc melting/suction casting	2–10	Fully amorphous	–	83	–	–	[41,42]
Zr_58.6_Al_15.4_Co_18.2_Cu_7.8_	Arc melting/suction casting	10	Fully amorphous	1950	84	–	–/2.0	[43]
Zr_55_Ti_3_Hf_x_Cu_32−x_Al_10_ (x = 0, 1, 2, 3, 4, 5)	Arc melting/suction casting	4–8	Fully amorphous	1695–1824	73–85	–	2.0–2.5/0–2.6	[44]
Zr_40_Ti_15_Cu_10_Ni_10_Be_25_	Arc melting/suction casting	3	Mainly amorphous	–	–	796	–	[45]
Zr_50_Ti_5_Cu_10_Ni_10_Be_25_	Arc melting/suction casting	3	Mainly amorphous	–	–	741	–	[45]
Zr_40_Ti_15_Cu_10_Ni_5_Si_5_Be_25_	Arc melting/suction casting	3	Partially amorphous	–	–	843	–	[45]
Zr_70_Ni_16_Cu_6_Al_8_	Arc melting/arc tilt casting	3	–	1500 *	70	–	2.2/0	[46]
Zr_65−x_Ti_x_Cu_20_Al_10_Fe_5_ (x = 0, 2, 4, 6, 8)	Arc melting/suction casting	2	Fully amorphous for x = 0, 2, 4, and partially amorphous for x = 6, 8	1405–1905	–	–	–/0–8.6	[47]
Zr_56_Cu_24_Al_9_Ni_7−x_Ti_4_Fe_x_ (x = 0, 1, 3, 5, 7)	Arc melting/suction casting	2	Fully amorphous for x = 0, 1, 3, and partially amorphous for x = 5, 7	1043–1709	–	–	3.9–6.3/0–5.6	[48]
Zr_60+x_Ti_2.5_Al_10_Fe_12.5-x_Cu_10_Ag_5_ (x = 0, 2.5, 5)	Arc melting/suction casting/casting	1–2	Fully amorphous	~1660–1740	70–78	443–460	2.0–2.0/4–12	[49]
Zr_55_Co_30_Ti_15_	Arc melting/melt spinning	0.04	Fully amorphous	–	–	–	–	[50]
Zr_62_Cu_22_Al_10_Fe_5_Dy_1_	Induction melting/melt spinning	0.04	Fully amorphous	–	96	495	–	[51]
Zr_37_Co_34_Cu_20_Ti_9_	Arc melting/melt spinning	–	Fully amorphous	–	81	567	–	[52]
Zr_40_Ti_35_Ni_14_Nb_11_	Magnetron co-sputtering	0.0006	Fully amorphous	–	122	~658	–	[53]
Zr_46_Ti_40_Ag_14_	Magnetron co-sputtering	0.0003	Fully amorphous	–	109	~567	–	[54]
Zr_46_Ti_43_Al_11_	Magnetron co-sputtering	0.0002	Fully amorphous	–	127	~520	–	[54]
Zr_62.5_Pd_37.5_	Magnetron sputtering	–	Fully amorphous	–	–	–	–	[55]

***** Determined in a tensile test.

**Table 2 jfb-13-00245-t002:** Recent exemplary Ti-based amorphous alloys considered for biomedical applications with their technological details and mechanical parameters: d—obtained diameter/thickness, σ_max_—compressive ultimate strength, E—Young’s modulus, *H*_V_—Vickers hardness, ε_el_—elastic strain.

Composition (at.%)	Production Method	d (mm)	Structure	σ_max_ (MPa)	E (GPa)	*H*_v_ (HV)	ε_el_/ε_pl_ (%)	Ref.
(Ti_55_Zr_15_Be_20_Ni_10_)_100−x_Fe_x_ (x = 0, 2, 4, 6, 8, 10)	Arc melting/suction casting	5–10	Fully amorphous	1878–2355	–	–	–/3.4–1.3	[127]
Ti_47_Cu_38_Zr_7.5_Fe_2.5_Sn_2_Si_1_Ag_2_	Arc melting/tilt pouring	7	Fully amorphous	2080	100	588	–/2.5	[128]
Ti_48_Cu_37_Zr_7.5_Fe_2.5_Sn_2_Si_1_Ag_2_	Arc melting/tilt pouring	6	Fully amorphous	2050	101	571	2.0/2.8	[129]
Ti_45_Zr_10_Cu_31_Pd_10_Sn_4_	Argon atomization/spark plasma sintering	6	Fully amorphous	–	100	–	–	[130]
Ti_47−x_Cu_40_Zr_7.5_Fe_2.5_Sn_2_Si_1_Sc_x_ (x = 0, 1, 2, 3, 4)	Arc melting/suction casting/tilt pouring	3–6	Fully amorphous	1982–2169	93–101	577–590	–/0.8–5.9	[131]
Ti_47_Cu_38−x_Zr_7.5_Fe_2.5_Sn_2_Si_1_Ag_2_Nb_x_ (x = 0, 1, 2)	Arc melting/suction casting	3–5	Fully amorphous	2031–2078	97–100	588–593	–/1.9–2.5	[132]
Ti_40_Zr_10_Cu_36_Pd_14_	Arc melting/tilt pouring	5	Fully amorphous	2010	96	556	2.0/0.7	[133,134]
Ti_40_Zr_10_Cu_36−x_Pd_14_Ga_x_ (x = 2, 4, 8, 10)	Arc melting/suction casting	3	Fully amorphous for x = 2, 4, and partially amorphous for x = 8, 10	1935–2075	93–140	–	1.9–2.1/0.8–2.5	[135]
Ti_47_Cu_40−x_Zr_7.5_Fe_2.5_Sn_2_Si_1_Ta_x_ (x = 1, 2, 3, 4)	Arc melting/suction casting	3	Fully amorphous	2041–2191	98–101	582–595	–/1.0–3.4	[136]
Ti_40_Zr_35_Cu_17_S_8_	Induction melting/arc melting/suction casting	3	Fully amorphous	3200	96	509	–	[137]
Ti_50_Zr_25_Cu_17_S_8_	Induction melting/arc melting/suction casting	2	Fully amorphous	3100	98	524	–	[137]
Ti_40_Zr_10_Cu_38_Pd_12_	Induction melting/mold casting	2	Fully amorphous	2300	95	734	–/4.0	[138]
Ti_40_Zr_10_Cu_34_Pd_14_Sn_2_	Arc melting/suction casting	1.5	Fully amorphous	>2000	93	–	2.2/–	[139]
Ti_60_Zr_15_Cu_17_S_8_	Induction melting/arc melting/suction casting	1	Fully amorphous	2800	98	547	–	[137]
TiCuZrPd:B_x_ (x = 0, 4, 8, 14)	Pulsed laser deposition	–	Fully amorphous	–	108–174	454–685	–	[140]
Ti_42_Zr_35_Ta_3_Si_5_Co_12.5_Sn_2.5_	Argon atomization/hot pressing	–	Fully amorphous	1261	79.7	–	–	[141]

## Data Availability

Not applicable.

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
