# Peer review of "(untitled)"

_jfb, 2022, doi:10.3390/jfb13040245_

Round 1
Reviewer 1 Report
Attached

Author Response
Dear Reviewer,
thank you very much for all your comments and suggestions, which significantly improve the quality of our article. Below you can find our responses (red) to your comments. We hope that our explanations are satisfactory. All remarks were considered according to your suggestions. Changes introduced into the manuscript are marked by the “Track changes” MS Word function.
Sincerely yours,
Mariusz Hasiak
-------------------------------------------
- In the introduction section, (sentence 29-34), the authors mention high Young’s modulus in various alloys including Zr and Ti based alloys causes stress shielding effect. Stainless steel and Co-Cr based alloys having high Young’s modulus (180-210 GPa) are prone to cause stress shielding effect. On the contrary, Kim et.al and Niinomi et.al have reported Ti and Zr based alloys with young’s modulus of 40-55 GPa and 14-25 GPa which are very close to that of bone (30-40 GPa). So, these alloys surpasses the effect of stress shielding in human bones. Justify your statement and mention the range of Youngs modulus in implants which is responsible to cause stress shielding effect in human body.
This is true that for Zr-based and Ti-based materials it is possible to obtain materials with very low Young modulus. This part of the paragraph was remodeled and rewritten to avoid generalization. Moreover, the Young modulus for the bones and the value of about 60 GPa which is already effective in eliminating the stress shielding effect according to Niinomi et al. are mentioned.
- Generally, due to excessive magnetic field interactions, ferromagnetic metallic implants have the risk of potential migration of implantsduring MRI.In introduction section, ( line 36-37), give examples of materials that are showing inapplicability for magnetic resonance and X-ray imaging.
The sentence was changed to give examples of inapplicable materials and incompatibility mechanisms – “Other problems relate to the inapplicability of some materials for magnetic resonance due to the displacement possibility (e.g. for ferromagnetic materials like ferritic and martensitic stainless steels [7]) or inducing severe artifacts (e.g. for austenitic stainless steels, Co-based alloys, and even Ti-based alloys [8]). Similarly, artifacts can be induced during X-ray imaging [9].”
- In lines (201-220), the author has discussed about the Zr based metallic glasses that are more prone to pitting corrosion in chloride containing solutions. This is true. And as a solution to this, the authors have stated that optimizing alloy composition by addition of alloying element like Ti is the only solution to it. In my knowledge, the corrosion behavior of amorphous alloys should be seen from the aspect of short and medium atomic ordering, amount of free volume, atomic clusters, crystalline inclusions and ultimately the chemical composition. The authors should take into consideration the others aspects also while drafting a review.
There is no statement in the manuscript that optimizing the chemical composition is the only solution to the vulnerability of Zr-based metallic glasses to pitting corrosion in chloride-containing environments. In fact, the mechanisms behind the enhanced corrosion resistance are concisely discussed in the introduction. The lack of grain boundaries and chemical composition homogeneity is mentioned in this section. The lack of crystalline inclusions and the common presence of alloying elements with a great ability to chemical passivation were also added. Moreover, the mentioned short and medium atomic ordering, amount of free volume, atomic clusters, and crystalline inclusions are closely related to chemical composition as well as chemical and structural homogeneity.
In the section about Zr-based metallic glasses corrosion, in addition to doping by varied alloying elements and compositional optimization, also other novel methods to improve the pitting corrosion resistance are mentioned. For example, the magnetron sputtering in the nitrogen presence, introducing the oxygen impurity to shrink the free volume highly active regions, thermal Ceramic Conversion Treatment (CCT), and femtosecond laser surface nano- and micro-structuring.
The beginning of the second paragraph in Section 2.2 about Zr-based metallic glasses corrosion resistance was changed to better indicate that the addition of the alloying elements is only one of the ways to improve the pitting corrosion resistance. Moreover, the ending of the penultimate paragraph of the same section was enriched with the description of free volume reduction during annealing and the worse performance of partially crystallized metallic glasses.
- Regarding the influence of the heat treatment, the biocompatibility and corrosion resistance do not change below Tg. Above Tg, metallic glasses lose their elasticity after crystallization due to the high stiffness and brittleness of the crystals.In lines (441-443), it is mentioned that “inflammation signs were observed in sample treated above 800 K, partially or fully crystallized confirming the positive influence of amorphous structure on biocompatibility”. Explain it.
This fragment was rewritten to make it clearer – “The other research on BALB/c mice was performed with the use of the as-cast and annealed Zr56Al16Co28 BMG samples [31]. For the samples in the as-cast state and after annealing below crystallization temperature no inflammatory response, and no cells dysplasia were revealed after 4 weeks since subcutaneous implantation in the dorsal region of mice. Only inflammation signs were observed for the samples heat-treated above the crystallization temperatures, so partially or fully crystallized. These results confirm the deterioration of cytocompatibility with a transition from amorphous to crystalline internal structure what can be associated with worse corrosion resistance and increased ions release.”
- While mentioning the in-vitro and in-vivo results, the ISO standards using which the test has been performed should be mentioned.
When possible, the standards used for in vitro, antibacterial, and in vivo studies were added. Mostly the ISO 10993 group of standards for in vitro studies, and JIS Z 2801 for antibacterial studies are in use. However much original research does not follow the particular standard, and it is not stated.
- In lines (680-693), in vitro cytocompatibility in Ti based alloys is given. It is shown that Ti based metallic glass alloys have shown better cytocompatibility in limited presence of alloying elements like Cu and Ag. How rise in percent of Cu and Ag is responsible for decrease in cytocompatibility should also be discussed.
In the manuscript, it is stated that the described alloys showed good cytocompatibility despite the presence of Cu and Ag in their chemical composition. This fragment was rewritten to make it clearer. The dose-dependent toxicity of Cu and Ag was described and discussed firstly in Section 2.3 (In vitro cellular studies for Zr-based materials) as Zr-based metallic glasses also contain Cu and Ag additions. Good cytocompatibility for metallic glasses with these two elements can be attained due to achievable low Cu and Ag ions release rates. This leads to not exceeding the cytotoxicity thresholds. In addition, the ions release is very dependent on the alloy's overall composition.
- In line (804-805), the authors have mentioned that metallic glasses with high Cu content exhibited no inflammatory signs and the implants were surrounded with new bone tissue. This indicates that Cu rich composition has also showed better osseointegration. This contradicts the results reported in in-vitro studies where it is shown that materials with high Cu percent are cytotoxic in nature. The authors should justify their statement.
The results of the original research performed by Lin et al. showed the good in vitro cytocompatibility of studied materials in contact with D1 mouse cells after 72 h of incubation. However, the tests of the corrosion medium after the materials' electrochemical corrosion examination showed slight cytotoxicity (78 % of viability) for the “high-Cu” metallic glass. It was associated with lower pitting corrosion resistance and resulting excessive ions release. The effects of these were not pronounced in in vivo tests, what can be caused by lower corrosion rates than in electrochemical test, and body fluid circulation removing the ions from the near-implant sites.
This paragraph was improved to state this more clearly and be more descriptive.
- The summary (3.6) should be merged with final summary given at the end. Conclusions should be written in separate section.
The adopted convention of this review article assumes separate summaries for the main parts of the manuscript (Zr-based materials and Ti-based materials). It was done to make the sections' summaries shorter and we believe that it makes the entire manuscript easier to follow. However, the changes were made to remove the final summary part and separate the rewritten Conclusions section.
Reviewer 2 Report
The paper summarizes and discusses the recent findings in the areas of mechanical properties, corrosion resistance, in vitro cellular research, antibacterial properties, and in vivo animal research of metallic glasses. It is well-written and organized. However, it requires the following major revision before publication.
1. The are so many challenges in the manufacturing approaches, size, phases, and bio-mechanical properties of metallic glasses, which significantly retard its application by clinicians and industries. The authors may discuss deeply these challenges and proposed some ways to address these challenges.
2. Authors may add some pictures to make the paper more appealing to readers.
Author Response
Dear Reviewer,
thank you very much for all your comments and suggestions, which significantly improve the quality of our article. Below you can find our responses (red) to your comments. We hope that our explanations are satisfactory. All remarks were considered according to your suggestions. Changes introduced into the manuscript are marked by the “Track changes” MS Word function.
Sincerely yours,
Mariusz Hasiak
------------------------------------------
- The are so many challenges in the manufacturing approaches, size, phases, and bio-mechanical properties of metallic glasses, which significantly retard its application by clinicians and industries. The authors may discuss deeply these challenges and proposed some ways to address these challenges.
Some of the challenges related strictly to discussed metallic glasses were already mentioned in the text. It especially includes the manufacturing size restriction as many metallic glasses are fabricated only as thin films or ribbons, and the bulk materials have a couple of millimeters in size. Also, other challenges related to some materials were mentioned like low glass forming ability, complicated manufacturing, lack of plasticity, brittleness, presence of possibly toxic elements, not ideal in vitro cytocompatibility or lacking pitting corrosion resistance.
However, our intention for this manuscript was to summarize and discuss the recent advances and research results in the field of metallic glasses for biomedical applications. The general challenges, connected with metallic glasses, which retards their application in the biomedical industry are an overly broad topic that is intended to be deeply covered in a separate review article.
- Authors may add some pictures to make the paper more appealing to readers.
According to the suggestion, a number of figures were added to the manuscript to make it more appealing to readers. It includes the following:
Figure 2. Stress-strain curves obtained in uniaxial compression test for the as-cast Zr56Cu24Al9Ni7-xTi4Fex (x = 0, 1, 3, 5, 7 at.%) alloys. Materials were fully amorphous for x = 0, 1, 3, and partially amorphous for x = 5, 7. (Reprinted with permission from ref. [48]. Copyright © 2022 Elsevier B.V.)
Figure 5. Polarization curves obtained in potentiodynamic test in 3.5 wt% NaCl environment for Zr61Ti2Cu25Al12 BMG with different oxygen impurity. (Reprinted with permission from ref. [69]. Copyright © 2021 Elsevier B.V.)
Figure 8. SEM images of human blood platelets on (a, d) bare glass, (b, e) Ti-coated glass, (c, f) glass coated with a thin film of Zr53Cu33Al9Ta5 metallic glass. Images (a–c) represent low magnification, and images (d–f) high magnification. (Reprinted with permission from ref. [84]. Copyright © 2018 Elsevier B.V.)
Figure 11. Images of needles retraction after puncturing the mouse dorsal skin for (a, b) bare stainless steel hypodermic needles, and (c, d) Zr53Cu33Al9Ta5 metallic glass coated hypodermic needles. (Reprinted from ref. [100] under Creative Commons CC-BY license)
Figure 12. Stress-strain curves for Ti40Zr10Cu36Pd14 BMG obtained in compression tests for samples with a diameter of 3 mm and a height of 5 mm. (Reprinted from ref. [136] under Creative Commons CC-BY license)
Figure 14 Wear rates of Ti40Zr10Cu38Pd12 BMG, and Ti-6Al-4V alloy measured in air, and phosphate buffered solution (PBS) corrosive environment. (Reprinted with permission from ref. [140]. Copyright © 2020 Elsevier B.V.)
Figure 16. Images of samples surface after the potentiodynamic tests in borate buffered 0.1 M NaCl solution for (a) Ti40Zr35Cu17S8BMG, (b) Ti50Zr25Cu17S8 BMG, (c) Cu47Ti34Zr11Ni8 BMG, and (d) Ti-6Al-4V alloy. (Reprinted with permission from ref. [139]. Copyright © 2019 Elsevier B.V.)
Figure 18. Ions release from Ti40Zr10Cu36Pd14 in artificial saliva solution. Measured by inductively coupled plasma mass spectrometry (ICP-MS) after 1, 3, and 7 days. (Reprinted and adapted from ref. [135] under Creative Commons CC-BY license)
Figure 21. Optical images showing the cross-section of rabbit bone with implanted porous Ti42Zr40Si15Ta3 metallic glass sample and bone ingrowths after six months in (a) low, and (b) high magnification. (Reprinted and adapted from ref. [167] under Creative Commons CC-BY license)
Figure 22. Relative to the control group survival rate of rabbit primary osteoblasts in extraction medium derived after different times from Mg60Zn35Ca5 BMG, Ti-6Al-4V alloy, and PLA polymer. (Reprinted and adapted from ref. [179] under Creative Commons CC-BY license)
Figure 23. (a, d) X-ray, (b, e) micro-computed tomography (μ-CT), and (c, f) 3D reconstruction images of rabbit bone defects filled with (a-c) Mg69Zn27Ca4 metallic glass, and (d-f) β-tricalcium phosphate (β-TCP) after two months since operation. (Reprinted with permission from ref. [178]. Copyright © 2019 Elsevier B.V.)
Figure 24. (a) Potentiodynamic test in Hank’s Balanced Saline Solution (HBSS) results for pure Ti, pure Ta, Ta57Ti17Zr15Si11, and Ta75Ti10Zr8Si7 metallic glasses; (b) Relative to control group D1 cells viability after 72 h obtained from MTS assay for pure Ti, pure Ta, Ta57Ti17Zr15Si11, and Ta75Ti10Zr8Si7 metallic glasses. (Reprinted with permission from ref. [183]. Copyright © 2018 Elsevier B.V.)
Figure 25. (a) Wear rates for Pd40Cu30Ni10P20 BMG, CoCrMo alloy, 316L stainless steel, and Ti-6Al-4V alloy in dry conditions and phosphate buffered saline (PBS) solution; (b) Polarization curves for Pd40Cu30Ni10P20 BMG, CoCrMo alloy, 316L stainless steel, and Ti-6Al-4V alloy obtained in potentiodynamic test in PBS. (Reprinted and adapted with permission from ref. [186]. Copyright © 2020 Elsevier B.V.)
Reviewer 3 Report
In the present review, the authors have made an attempt to showcase the use of metallic glass for biomedical applications. However, there are other reviews on the similar subject. The language and coherency undermine the promise of the manuscript.
Abstract: The first line is too long and needs to be divided into 2 sentences. "This review article synthesizes 13 the latest results with a theoretical basis to explain them." What does this line mean? cellular research or cellular studies? Likewise, the same for animal studies.
Introduction:
"All these effects lower their applicational biocompatibility defined as the ability to induce no measurable harm to the host" should be changed and re written. Line 40: break imposed limitations or overcome imposed limitations?
The introduction is not fluid and does not give the gist of the intended manuscript. What is the reason for including the number of publications since 1970 onwards?
“What especially should be considered are reported in the literature values of achievable elastic strain, which are higher than 2 % in every case and reach even astonishing, as for metallic material, value of 6.28 % for the Zr56Cu24Al9Ni7Ti4 BMG” does not sound correct.
Line 130 astonishing word does not come into the purview of terms used for scientific literature.
Line 151 is not fluid and does not make sense.
Line 207 Indefinite sentence
Line 220 instead of how important please change to the importance
Please give the abbreviation of SAOS-2. In some places it is saos-2, please maintain uniformity.
Line 359: Can also “be” improved
Line 399 99.999% lethality? Please keep the decimal place to one- or two-digit values only.
Line 411 Please check that the figure legend should be on the similar page
Line415 from the “available” literature
Line 460 in vivo to be italicized
Line 521 “That have gained” and not gaining attention are
Line 529 The sentence “Ti became also of the most used” has to be changed
Line 794 is incomplete after Sprague-Dawley rats?
Line 868 recently, usually in the same line?
Line 946 Please remove the phrase “what is more”
Line 962 Please remove the phrase “what is more”
The abstract, introduction and conclusion is very discordant. It is very confusing to the readers. The lacunae for writing the review are not highlighted in the abstract or the introduction. The wealth of superfluous words should be limited. The discussion regarding platinum based metallic glass has not found any mention. The manuscript lacks vision and the author fails to convey the true essence of the manuscript.
I find the conclusion to be non-justifiable with no major focus on the future consideration.
Author Response
Dear Reviewer,
thank you very much for all your comments and suggestions, which significantly improve the quality of our article. Below you can find our responses (red) to your comments. We hope that our explanations are satisfactory. All remarks were considered according to your suggestions. Changes introduced into the manuscript are marked by the “Track changes” MS Word function.
Sincerely yours,
Mariusz Hasiak
------------------------------------------
- There are other reviews on the similar subject. The language and coherency undermine the promise of the manuscript.
The ongoing rapid development of metallic glasses for biomedical applications causes the quick obsolescence of existing older reviews. The lack of an overview of the latest research results, solved problems, and emerging issues justifies the need to summarize the recent advances in the field. It is also a response to the current high interest in the topic of biocompatible materials including advanced metallic glasses which have the potential to improve current treatment results.
The text of the article has been rechecked several times to improve the language and eliminate English errors.
- Abstract: The first line is too long and needs to be divided into 2 sentences. "This review article synthesizes 13 the latest results with a theoretical basis to explain them." What does this line mean? cellular research or cellular studies? Likewise, the same for animal studies.
The first line of the abstract was divided into two sentences: “The continuous development of novel materials for biomedical applications is resulting in an increasingly better prognosis for patients. Application of more advanced materials relates to fewer complications and a desirable higher percentage of successful treatments.”.
The sentence “This review article synthesizes the latest results with a theoretical basis to explain them.” was rewritten to “This review article intends to synthesize the latest research results in the field of biocompatible metallic glasses to create a more coherent picture of these materials” to improve the clarity.
According to the suggestion “cellular research” and “animal research” were changed to “cellular studies” and “animal studies” respectively.
- "All these effects lower their applicational biocompatibility defined as the ability to induce no measurable harm to the host" should be changed and re written. Line 40: break imposed limitations or overcome imposed limitations?
The sentence was rewritten as “Ultimately, the mentioned issues lower the applicational biocompatibility of current materials for biomedical applications”. The definition of biocompatibility was moved to the beginning of the manuscript. The “break imposed limitations” was changed to indeed better “overcome imposed limitations”.
- The introduction is not fluid and does not give the gist of the intended manuscript. What is the reason for including the number of publications since 1970 onwards?
The introduction was moderately remodeled to make it more fluent. Its purpose was to mention why metallic glasses are considered for biomedical applications, show the historical view on the beginnings of these materials, and generally present this materials group along with its properties to the reader. It implies the use a couple of historicaly valuable publications.
- “What especially should be considered are reported in the literature values of achievable elastic strain, which are higher than 2 % in every case and reach even astonishing, as for metallic material, value of 6.28 % for the Zr56Cu24Al9Ni7Ti4 BMG” does not sound correct.
This sentence was rewritten into three separate sentences: “Noteworthy are reported in the literature high values of elastic strain achievable for Zr-based metallic glasses. This parameter is higher than 2 % for each material from Table 1 for which data are available. The elastic strain can reach even remarkable, as for metallic material, value of 6.28 % for the Zr56Cu24Al9Ni7Ti4 BMG”.
- Line 130 astonishing word does not come into the purview of terms used for scientific literature.
The word “astonishing” was replaced by “remarkable”.
- Line 151 is not fluid and does not make sense.
The entire paragraph was remodeled to make it clearer and more fluent. The definition of the stress shielding effect was moved to the Introduction section.
- Line 207 Indefinite sentence
The sentence was corrected to indicate the subject.
- Line 220 instead of how important please change to the importance
The sentences were rewritten with the use of “importance” to be correct and clearer.
- Please give the abbreviation of SAOS-2. In some places it is saos-2, please maintain uniformity.
The Saos-2 is not a strict abbreviation but a human osteogenic sarcoma cell line designation. At the first use, the “[…] Saos-2 human osteosarcoma (sarcoma osteogenic) cells […]” explanation was added. Moreover, the correction was made to use the Saos-2 form in all places.
- Line 359: Can also “be” improved
The order of words was corrected.
- Line 399 99.999% lethality? Please keep the decimal place to one- or two-digit values only.
The value of 99.999 % is reported in the original article Lee, J.; Liou, M.-L.; Duh, J.-G. The Development of a Zr-Cu-Al-Ag-N Thin Film Metallic Glass Coating in Pursuit of Improved Mechanical, Corrosion, and Anti-microbial Property for Bio-Medical Application. Surf Coat Technol 2017, 310, 214–222, doi:10.1016/j.surfcoat.2016.12.076. We wanted to keep the precision that was achieved in the original research.
- Line 411 Please check that the figure legend should be on the similar page
The “for peer review” version of the manuscript is yet before final editing and formatting which is carried out by the publisher. Therefore, this manuscript version was not formatted visually by us.
- Line415 from the “available” literature
“Literature” word was added to the text.
- Line 460 in vivo to be italicized
In vivo is now italicized there.
- Line 521 “That have gained” and not gaining attention are
This sentence was rewritten to “Ti-based materials are another group of metallic glasses that has recently gained much attention”.
- Line 529 The sentence “Ti became also of the most used” has to be changed
This part of the sentence was changed to “[…] Ti became one of the most widespread and beneficial elements in their compositional design […]”.
- Line 794 is incomplete after Sprague-Dawley rats?
The unnecessary “with” phrase was removed to make the sentence complete.
- Line 868 recently, usually in the same line?
The sentence was rewritten to “Recently studied Mg-based metallic glasses mainly belong to MgZnCa alloys […]”.
- Line 946 Please remove the phrase “what is more”
The “What is more” phrase was removed.
- Line 962 Please remove the phrase “what is more”
The “What is more” phrase was removed.
- The abstract, introduction and conclusion is very discordant. It is very confusing to the readers. The lacunae for writing the review are not highlighted in the abstract or the introduction. The wealth of superfluous words should be limited. The discussion regarding platinum based metallic glass has not found any mention. The manuscript lacks vision and the author fails to convey the true essence of the manuscript.
Taking into consideration the rapid development of biocompatible metallic glasses the intention of our review was to sum up, compare and discuss the recent advances and research results in this field. It was done to provide an opportunity to get knowledge of the current state of research and direct future studies. The abstract and the end of the introduction were changed to highlight it more clearly and be more in line with the conclusions. Moreover, the conclusions were rewritten to be more specific and address the mentioned issues.
In our manuscript, we focused only on palladium-based metallic glasses. The brief discussion that concerned the novel discoveries in this field was included in the final summary and rewritten conclusions.
- I find the conclusion to be non-justifiable with no major focus on the future consideration.
The Conclusions section was rewritten to better capture the essence of the article. It was significantly expanded to put greater emphasis on future considerations.
Round 2
Reviewer 2 Report
Issues addressed by the authors. I have no more concerns.
Reviewer 3 Report
I appreciate the authors for the modifications as per the suggestions for "Review on biocompatibility and prospect biomedical applications of novel functional metallic glasses".
Overall, the content has been improved. I would like the authors to check for minor grammar errors in the manuscript.